# RE-ALIGNING LANGUAGE TO VISUAL OBJECTS WITH AN AGENTIC WORKFLOW

**Yuming Chen**[1]    **Jiangyan Feng**[2]    **Haodong Zhang**[2]    **Lijun Gong**[2]    **Feng Zhu**[2]
**Rui Zhao**[2]    **Qibin Hou**[1*]    **Ming-Ming Cheng**[1]    **Yibing Song**[*]

[1]VCIP, Nankai University    [2]SenseTime Research

chenyuming@mail.nankai.edu.cn    houqb@nankai.edu.cn    yibingsong.cv@gmail.com

## ABSTRACT

Language-based object detection (LOD) aims to align visual objects with language expressions. A large amount of paired data is utilized to improve LOD model generalizations. During the training process, recent studies leverage vision-language models (VLMs) to automatically generate human-like expressions for visual objects, facilitating training data scaling up. In this process, we observe that VLM hallucinations bring inaccurate object descriptions (*e.g.,* object name, color, and shape) to deteriorate VL alignment quality. To reduce VLM hallucinations, we propose an agentic workflow controlled by a large language model (LLM) to re-align language to visual objects via adaptively adjusting image and text prompts. We name this workflow Real-LOD, which includes planning, tool use, and reflection steps. Given an image with detected objects and VLM raw language expressions, Real-LOD reasons its state automatically and arranges action based on our neural symbolic designs (*i.e.,* planning). The action will adaptively adjust the image and text prompts, and send them to VLMs for object re-description (*i.e.,* tool use). Then, we use another LLM to analyze these refined expressions for feedback (*i.e.,* reflection). These steps are conducted in a cyclic form to gradually improve language descriptions for re-aligning to visual objects. We construct a dataset that contains a tiny amount of 0.18M images with re-aligned language expression and train a prevalent LOD model to surpass existing LOD methods by around 50% on the standard benchmarks. With automatic VL refinement, our Real-LOD workflow reveals a potential to preserve data quality along with scaling up data quantity, further improving LOD performance from a data-alignment perspective.

## 1 INTRODUCTION

Aligning language expressions with visual objects has been continuously evolving. Initially, a single noun word is used as a category label (Redmon et al., 2016; Ren et al., 2016; Carion et al., 2020) to connect a visual object. Then, phrases are introduced (Akbari et al., 2019; Li et al., 2022; Gao et al., 2023) to describe objects. Further, referring expressions (Su et al., 2020; Zhang et al., 2022) and complete descriptions (Schulter et al., 2023; Yao et al., 2024) are developed for object detection. Although language evolves from coarse labels to fine-grained expressions, the essence of object detection is to align the language data to visual objects. This alignment is challenging as language expressions become diverse to represent various human intentions. As for the same visual object, different people usually describe it in various forms, as they focus on different aspects of object properties (*e.g.,* color, shape, texture, and relationship with surroundings). This diversity makes vision language (VL) alignment cumbersome, where a comprehensive set of language expressions should be collected for model training. Fortunately, emerging VLMs (Zhang et al., 2021; Liu et al., 2023a; Ye et al., 2023; Sun et al., 2024; Yuan et al., 2024; You et al., 2024; Zhang et al., 2024) have recently been leveraged to produce human-like expressions. The auto-generation of language expressions for visual objects eases the difficulty of collecting training data pairs. By training LOD models with more VL data, studies (Pi et al., 2024; Dang et al., 2024; Kong et al., 2024) improve detection performance accordingly, especially when the language query is diverse to describe the target object.

---

[*]Q. Hou and Y. Song are corresponding authors. J. Feng and H. Zhang are equal contribution. The code is available at https://github.com/FishAndWasabi/RealLOD.

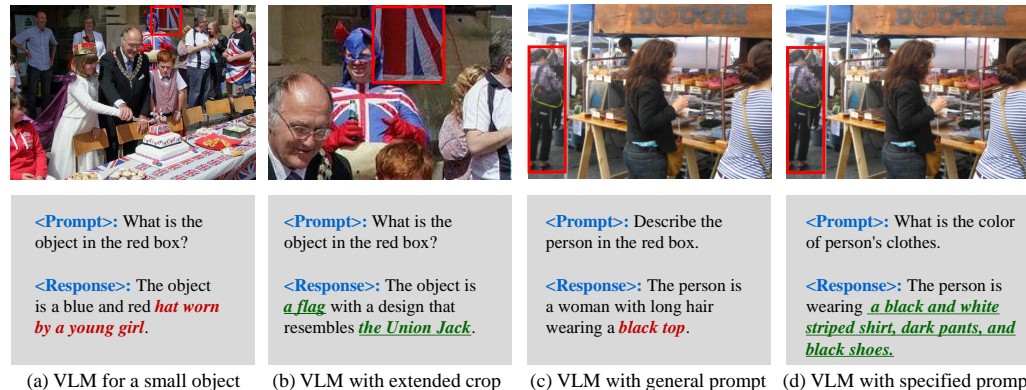

| (a) VLM for a small object | (b) VLM with extended crop | (c) VLM with general prompt | (d) VLM with specified prompt |

Figure 1: Examples of adaptive image and prompt modifications refine language expressions. For a small object in (a), VLM produces erroneous content marked in red. In (b), we crop the local region of (a) and obtain refined content marked in green. Another example is in (c), where a general prompt leads to erroneous content while a specific prompt in (d) does not.

The language expressions generated via VLMs, although aligned with human preference, may not accurately describe the target object due to model hallucinations. Fig. 1 shows two examples. A small object shown in Fig. 1(a) leads VLMs to generate erroneous expressions. Moreover, a general text prompt without specifying the target object shown in Fig. 1(c) makes VLMs incorrectly describe visual content. For model hallucinations on small objects, we analyze that VLMs are trained via extensive image-caption pair data (Radford et al., 2021; Schuhmann et al., 2022), where the caption mainly depicts global image content rather than local objects. The ignorance of local object context in training data makes VLMs hallucinate small objects. On the other hand, text prompts without specifying the target object (*e.g.,* 'in a red box') lead to incorrect detail descriptions from VLMs. A lack of object identity in the prompt makes VLMs insensitive to object details and expresses them erroneously. When adding inaccurate language expressions, the alignment of object and language becomes fragile and impedes LOD performance improvement along with VL data scaling up.

In this work, we propose to re-align language expressions to visual objects automatically to refine VL data quality from the alignment perspective. Our re-alignment is conducted via a workflow controlled by an LLM-powered agent (*i.e.,* Real-Agent).[1] Fig. 2 shows a glimpse of where there are three steps (*i.e.,* planning, tool use, and reflection) to form a cycle. Given an input image with detected objects, we first convert this image into captions, which are sent into Real-LOD together with object location, category, and raw language expressions from the VLMs initially used. Then, our agent automatically reasons the current

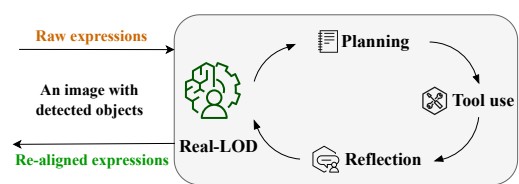

Figure 2: Glimpse of our Real-LOD. It takes image captions with detected objects and raw expressions as inputs. It gradually re-aligns expressions to match objects well. By using better-aligned training data pairs, we improve the performance of LOD.

state and arranges further actions. The state/action represents our neural symbolic design in the workflow, where we have predefined five states indicating how language aligns with the visual objects. Each state is followed by an arranged action. After the planning step, our agent takes action to construct adaptive VL prompts for the tool models (*i.e.,* VLM/LLM). Customized prompts enable tool models to collect more visual observations or refine current expressions. After the tool use step, the refined expression is sent to an LLM-based reflector for feedback. The feedback is then provided to our agent for planning in the next cycle. Fig. 5 shows an example in which the raw expression is gradually refined to align with the target object.

---

[1]For presentation clarity, we refer to Real-LOD as our agentic workflow, Real-Agent as the LLM-powered agent, Real-Data as our constructed dataset, and Real-Model as our trained LOD model.

Our Real-LOD refines language-object data pairs via re-alignment for LOD model training. Our Real-Model is a prevalent model structure with a Swin-B backbone (Liu et al., 2021). We train this model using our constructed dataset Real-Data, where there are 0.18M images that contain 1.4M language-object paired data. In the standard benchmarks (Mao et al., 2016; Schulter et al., 2023; Xie et al., 2023), we surpass existing methods by around 50%. This indicates that data quantity and quality are important for LOD training. In addition to the amount of image data that scales up, our Real-LOD can preserve the quality of the data pair. This potential directs a new trend that expanding high-quality paired data further improves LOD performance from a data-alignment perspective.

## 2 RELATED WORK

**Language-based object detection.** LOD requires models to locate the associated instances according to diverse expressions. Benefiting from visual-language detector development, the accuracy of LOD tasks is improved rapidly. MDETR (Kamath et al., 2021) first proposes an end-to-end modulated detector that detects objects by a given query. GLIP (Li et al., 2022) presents a language-image pre-train model for understanding object-level, category-aware visual representations. GDINO (Liu et al., 2024b) introduces an open-set object detector within an effective fusion module that allows the detection of objects with textual inputs such as category names or referring expressions. FIBER (Dou et al., 2022) designs a new visual-language model architecture that can handle different tasks such as visual question answering (VQA), image caption, object detection, and so on. APE (Shen et al., 2024) introduces a universal visual perception model to align visual and language representation on broad data at once so that it can conduct different language-visual tasks without task-specific fine-tuning. OWL-V2 (Minderer et al., 2023) proposes an architecture without any fusion modules. They use 1B language-object pair data to align image and textual features directly. The above methods utilizes language-object pair data to train their detectors, including COCO (Lin et al., 2014), Objects365 (Shao et al., 2019), OpenImage (Kuznetsova et al., 2020), SBU (Ordonez et al., 2011), GoldG (Li et al., 2022), CC (Sharma et al., 2018; Changpinyo et al., 2021; Xu et al., 2023), LVIS (Gupta et al., 2019), Flickr30K (Plummer et al., 2017), GRIT (Gupta et al., 2022), and V3Det (Wang et al., 2023a).

**Agentic workflows.** Intelligent agents empowered by LLMs are able to solve a wide range of complex tasks by following user's instructions (Askell et al., 2021; Liu et al., 2025; Significant Gravitas, 2023; Yohei Nakajima, 2023; Reworkd, 2023). Due to the strong understanding and reasoning abilities of LLM (Wei et al., 2022; Wang et al., 2023b), these agents are capable of making plans to achieve specified goals, mastering tools to execute tasks (Yao et al., 2023; Liu et al., 2023b; Tang et al., 2024; Yang et al., 2023a; Guo et al., 2024; Shen et al., 2023; Cai et al., 2024), generating reflection to refine outputs (Madaan et al., 2023; Shinn et al., 2023; Yu et al., 2024; An et al., 2023; Gou et al., 2024), and even collaborating with other agents (Chen et al., 2025; Xu et al., 2024; Holt et al., 2024). HuggingGPT (Shen et al., 2023) is presented as a powerful agent that leverages LLM to connect various AI models for solving different tasks. This agent is designed to understand and dismantle given AI tasks, as well as plan and select appropriate AI models to execute each subtask automatically. Similarly, LLaVA-Plus (Liu et al., 2023b) maintains a skill repository that contains a wide range of vision-language tools to fulfil many real-world multi-modal tasks. Other examples include Gorilla (Patil et al., 2023), GPT4tools (Yang et al., 2023a), and ToolAlpaca (Guo et al., 2024), which are fine-tuned LLMs with the ability to utilize available APIs. Additionally, recent studies have also shed light on improving agent performance through train-free approaches. One of the main ideas is reflection, where agents provide feedback to themselves and use it to refine their outputs. Self-Refine (Madaan et al., 2023) and Reflexion (Shinn et al., 2023) are the typical examples to reinforce agents with linguistic feedback, while CRITIC (Gou et al., 2024) introduces external tools into the reflection process with a human-predefined execution logic which is relatively fixed. Different from previous works, Real-LOD pioneerly designs an entire agentic workflow containing the above three steps to advance the alignment quality of VL data for LOD tasks.

## 3 RE-ALIGNING LANGUAGE TO VISUAL OBJECTS

In this section, we first revisit the LOD framework, showing how paired VL-inputs predict target objects and previous methods to generate language expressions in Sec. 3.1. Then, we illustrate the key steps of our Real-LOD (*i.e.,* planning, tool use, reflection) in Sec. 3.2. An example is provided in Fig. 5 to intuitively demonstrate how language expression is refined via our re-alignment scheme.

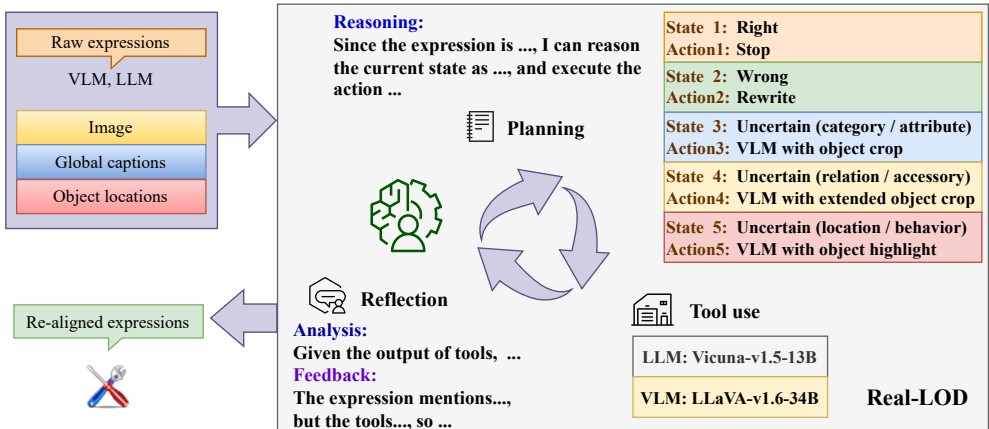

Figure 4: Overview of our agentic workflow. The inputs are images with captions, detected objects, and raw expressions. Our Real-Agent reasons the state and arranges the action (*i.e.,* planning). During action execution, our Real-Agent uses VLM and LLM to re-perceive visual content and refine expressions (*i.e.,* tool use). Then, the output results are analyzed by an LLM (*i.e.,* reflection). The feedback is provided to Real-Agent for planning in the next cycle.

In Sec. 3.3, we also analyze the refined expressions, which constitute training data pairs to improve LOD performance.

## 3.1 LOD FRAMEWORK AND LANGUAGE EXPRESSION GENERATIONS

The language-based object detection (LOD) framework typically consists of two encoders, a few interaction modules, and a decoder. Fig. 3 shows an overview. The inputs of LOD are one image and language expressions formulated by words, phrases, or sentences. LOD uses image and text encoders to obtain their embeddings independently. Then, the expressions interact with visual objects to formulate a joint cross-modal feature space. These interactions are usually conducted via cross-attention operations. Afterwards, LOD introduces a decoder module to localize the corresponding object based on each expression. The training losses (*e.g.,* L1 loss, GIOU loss (Rezatofighi et al., 2019), contrastive loss (Li et al., 2022)) are typically from DETR-based methods (Carion et al., 2020; Kamath et al., 2021; Zhang et al., 2023).

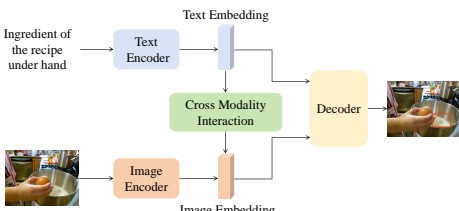

Figure 3: Overview of a general LOD framework. The paired VL data are independently encoded and then interacted to decode results.

The LOD framework establishes the connection between language and objects. The training data contains images, object bounding boxes (bbxs), and language expressions. Previous datasets (Mao et al., 2016; Plummer et al., 2017; Krishna et al., 2017) tend to collect expressions from human participants, which constructs a limited amount of paired data and bottlenecks the detection performance. Recently, studies (Dang et al., 2024; Pi et al., 2024) have leveraged VLMs to generate human-like expressions for visual objects. The training data amount is extensively scaled up, and the learned LOD model captures diversified object descriptions. Following their spirit, we use a VLM model, LLaVA-v1.6-34B (Liu et al., 2024a), to generate $673k$ language expressions for $188k$ images with $336.5k$ objects. Also, we use an LLM model, Vicuna-v1.5-13B (Zheng et al., 2023), to expand the number of expressions from $673k$ to $1,346.1k$ by generating synonyms. The details of raw expression generation are presented in Sec. A of the Appendix. After obtaining language-object paired data, we use SigLIP (Zhai et al., 2023) to calculate the VL matching score. For the paired data whose score is lower than 0.5, we use Real-LOD to re-align raw language expressions as illustrated in Sec. 3.2. This is because we leverage SigLIP to exclude about 75% of training data from our workflow, leaving only nearly 25% to be processed.

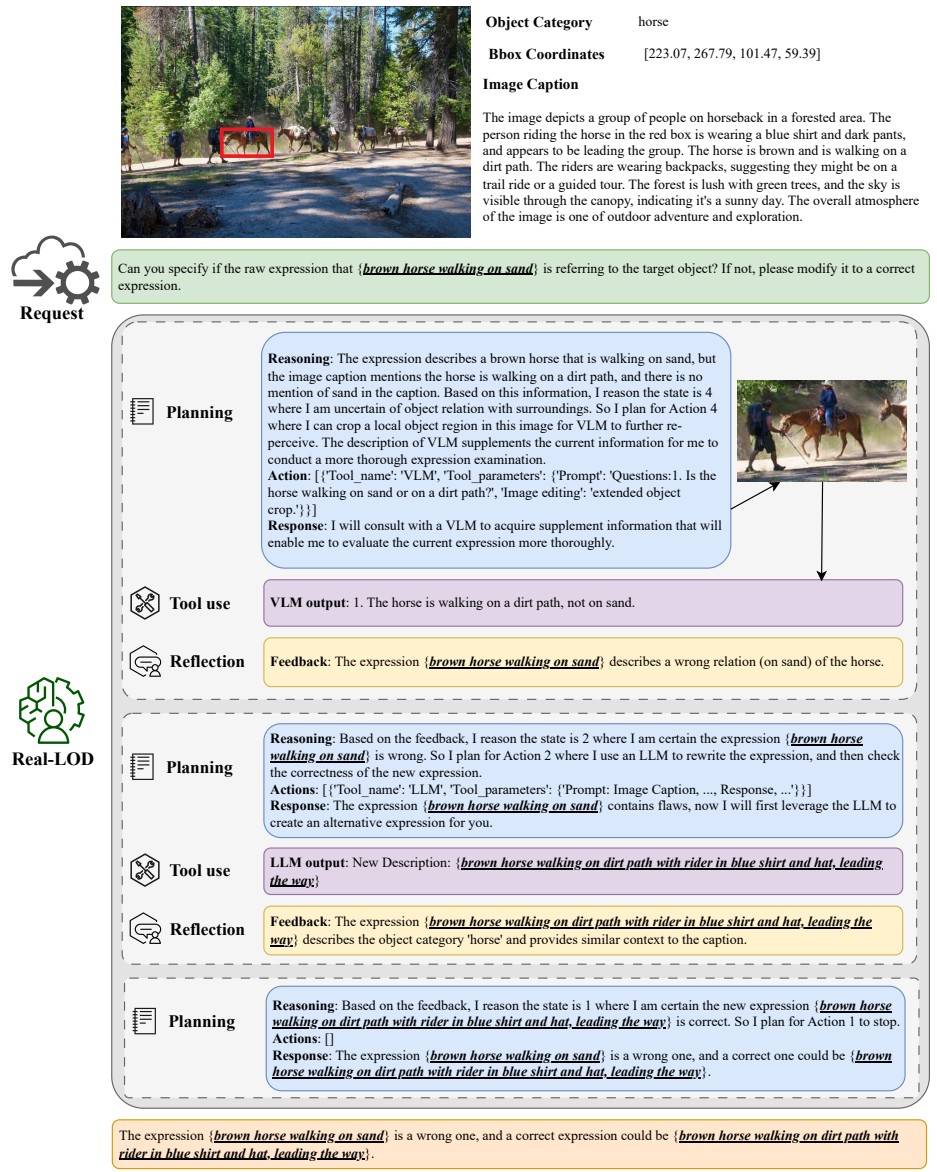

Figure 5: An example of how Real-LOD re-aligns one raw expression to the given image. Based on the input image, caption, and detected objects, Real-LOD performs planning, tool use, and reflection in a cyclic workflow for state reasoning, action execution, and result feedback. The image and prompt are adaptively adjusted for tool models to supplement customized object descriptions, which benefit expression re-alignment.

## 3.2 AGENTIC WORKFLOWS FOR LANGUAGE EXPRESSION RE-ALIGNMENT

The generated language expressions may not match visual objects. As illustrated in Sec. 1, either local context ignorance or unspecific text prompts lead to model hallucination. To solve this problem, we design a cyclic workflow to enable VLM to adaptively focus on local regions and specify text prompts according to the target object. Based on the finding that an LLM reasons more accurately in pure language form than in VL form, we choose a fine-tuned ChatGLM-6B (Zeng et al., 2023) with text-only input as our Real-Agent to control this workflow. Fig. 4 shows an overview. It consists of planning, tool use, and reflection steps to gradually refine raw expressions. To facilitate VL re-alignment, we have performed neural symbolic design in the planning and tool use steps where we predefined 5 states and actions, which are illustrated as follows.

**Planning.** In this step, we have predefined 5 states indicating how expressions are aligned to the target object from the view of VLM. Each state corresponds to one action to be executed. Formulating these states/actions is motivated via our data analysis in Sec. 3.3, where we observe how VL misalignment occurs in practice. Given the text-only input containing the expression, image caption, object category, and reflector output from the last cycle (Empty if at first cycle), our LLM-powered Real-Agent reasons the current state and arranges action accordingly. The five predefined states/actions are as follows:

`State 1: Right. Action 1: Stop.` Real-Agent is certain that the current language expression matches the target object. Real-Agent will stop the workflow and output the current expression.

`State 2: Wrong. Action 2: Rewrite.` Real-Agent is certain that the current expression does not match the target object. Hence, Real-Agent will use an LLM to regenerate the expression. The in-context prompt for rewriting will be generated following the template in Tab. 9 of the appendix for rewriting.

`State 3: Uncertain (category/attribute). Action 3: VLM with object crop.` Real-Agent is uncertain whether the current expression matches the target object. The uncertainty resides in the object category or attribute. So Real-Agent plans to crop the object region and use a VLM for further re-perception. The description from VLM will be kept in the text prompt for the next step.

`State 4: Uncertain (relation/accessory). Action 4: VLM with extended object crop.` Similar to State 3, Real-Agent is uncertain of object relation (with surroundings) or accessory. It plans to crop a larger region covering the target object and uses a VLM for re-perception. The description from VLM will be kept in the text prompt for the next step.

`State 5: Uncertain (location/behavior). Action 5: VLM with object highlight.` Similar to State 3, Real-Agent is uncertain of object location (in image) or behavior. It plans to highlight the object region using a red rectangle (Shtedritski et al., 2023) and uses a VLM for re-perception. The description from VLM will be kept in the text prompt.

When executing actions, we only refine language expressions in Action 2, while in Actions 3,4,5, we use VLM to supplement descriptions as in-context prompts. These prompts will be utilized in the next cycle to facilitate state reasoning and action executions.

**Tool use.** In the planning step, Real-Agent has scheduled to use several tools (*i.e.,* VLM and LLM) when executing actions. We prepare a toolset in advance where there is one LLaVA-v1.6-34B model for VLM usage and one Vicuna-v1.5-13B model for LLM usage. Based on its reasoning about the state of the current expression, Real-Agent adaptively modulates visual content and text prompts by setting up "Prompt" and "Image editing" parameters for scheduled tools. Then, the tool can be used effectively to get desired responses from VLM to improve the expression refinement. For example, when executing VLM for visual content re-perception, Real-Agent will edit the image via cropping or highlighting as planned according to the object bbxs. In addition, the customized text prompts designed by Real-Agent are more specifically related to the target object. In this way, Real-LOD can effectively reduce model hallucinations, improving language and object connections by re-aligning expressions. The visual and language prompts for VLM are shown in Tab. 12 of the appendix.

**Reflection.** After using tools, Real-Agent has finished action executions. We use an LLM (*i.e.,* Vicuna-v1.5-13B) as a reflector to analyze the results by incorporating the image caption. It verifies whether the current expression matches the target object. For State 3-5, where Real-Agent is uncertain, the reflector helps Real-Agent be confident in judging whether the expression is correct or wrong. For State 2 where Real-Agent has planned to rewrite the expression, the reflector examines the correctness of the new expression. The analysis of the reflector will be formulated as feedback to Real-Agent to facilitate its planning in the next cycle.

## 3.3 DATA ANALYSIS OF LANGUAGE AND VISUAL OBJECTS

**Training data for Real-Agent.** We prepare training data in the text form to fine-tune Real-Agent from ChatGLM-6B. First, we randomly collect images with detected objects from Objects365 (Shao et al., 2019) datasets. Then, similar to Sec. 3.1, we use VLM to generate raw expressions and collect the data pairs that are filtered out by SigLIP, *i.e.,* the matching score is lower than 0.5. In total, we prepare 15k input data to train Real-Agent. Each input data contains the object category, raw expression, and reasoning from the LLM-based reflector defined in Sec. 3.2 to examine whether the raw expression

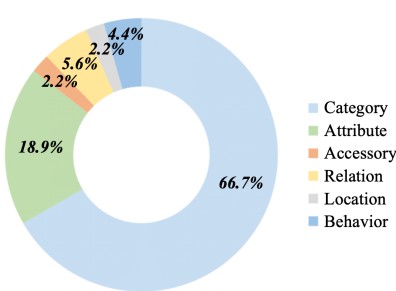

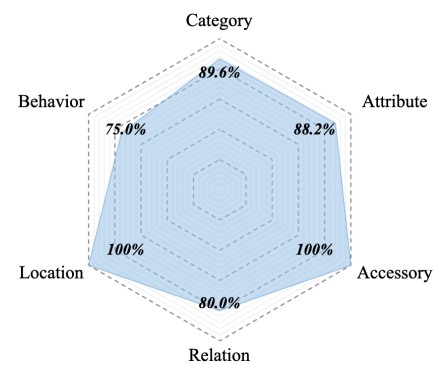

Figure 6: Percentage of 6 aspects for mismatch expressions where category and attribute consume the majority.

Figure 7: Success rate of expression re-alignment via Real-LOD in 6 aspects.

matches the target object. Then, we manually set the state for each input data by ourselves and collect responses (including "reasoning" and "actions") via an LLM (*i.e.,* Vicuna-v1.5-13B) with text prompts including several hand-crafted in-context examples in Tab. 11 following the spirit of LLaVA (Liu et al., 2023a). Finally, a manual check is conducted to ensure no error in the fine-tuning data. The training process is conducted in a parameter-efficient form, *i.e.,* LoRA (Hu et al., 2021), that does not affect the reasoning ability of ChatGLM-6B.

**Analysis of language and visual objects.** Our Real-LOD corrects raw expressions filtered out via SigLIP. As we design actions for expression correction in advance, we analyze how these expressions misalign to the target object. We randomly select three hundred filtered expressions and manually check each for a detailed observation. Overall, we summarize the misalignment reasons in 6 aspects based on the observed expressions: 1) **Category**: the expression describes another object rather than the target one; 2) **Attribute**: the expression provides wrong attributes such as color, shape, and texture of the target object; 3) **Accessory**: incorrect accessory descriptions of the target object; 4) **Location**: wrong relative location of the target object in the image; 5) **Relation**: incorrect object relationship with surroundings; 6) **Behavior**:

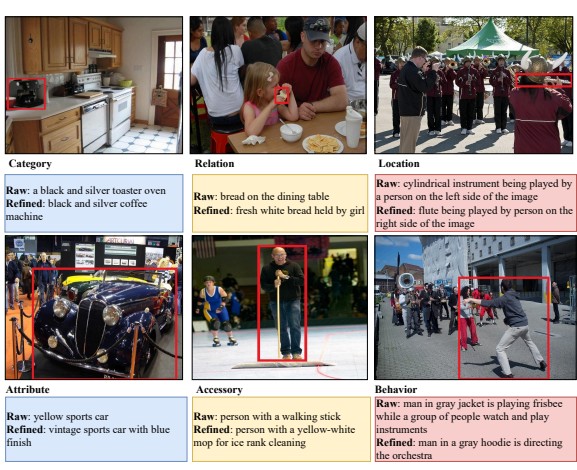

Figure 8: Example summary of misaligned raw expressions in 6 aspects, followed by our re-alignment.

incorrect object/human behaviors. Fig. 8 shows representative examples of how expressions listed in each aspect will be corrected. These 6 aspects motivate our neural symbolic action designs in Sec. 3.2 where image editing operations (*i.e.,* 'object crop', 'extended object crop', and 'object highlight') are utilized for VLM to perceive object related contents further. Fig. 6 shows the percentage of these 6 aspects in our observed expressions where most of them reside in the category and attribute aspects. After refinement, we compute the success rate of expressions from each aspect as shown in Fig. 7, where major expressions from category and attribute aspects can be effectively refined.

**Analytical experiment for Real-LOD.** Besides summarizing 6 aspects of mismatched raw expressions, we analyze how effective our Real-LOD is for re-alignment. Although we have designed corresponding actions to enable VLM for a re-perception, the accurate state reasoning and action planning will determine the refinement quality. For the inputs listed in Fig. 4, Real-Agent shall reason accurately to identify which state they belong to and execute the action accordingly. To analyze the reasoning ability of Real-Agent, we sample $11k$ samples and introduce a scheme for comparison by replacing the planning step with a step where one of the states/actions is selected randomly for further expression refinement. The reflector is used in both workflows to identify whether the final

expression refinement is successful, and we set the maximum round to 3. After refining expressions using Real-LOD and the random selection, we find that the success rate [2] is 74.7% v.s. 35.6%. This comparison shows that accurate reasoning from Real-Agent significantly improves expression correctness. Furthermore, we examine the matching score between image and expressions refined by these two refinement schemes via SigLIP. Our Real-Agent improves the average matching score by 66.27% (*i.e.,* from 0.0673 to 0.1119), while random selection improves by 32.69% (*i.e.,* from 0.0673 to 0.0893). This indicates Real-Agent improves the SigLIP matching score more than the random selection scheme (*i.e.,* 66.27% v.s. 32.69%). From the comparisons of re-alignment success rate and SigLIP score improvement, our Real-Agent demonstrates effectiveness in reasoning input state, planning action correctly, and successfully refining raw expressions. We also provide more analytical experiments in Sec. K.

## 4    EXPERIMENTS ON LANGUAGE-BASED OBJECT DETECTION

Real-Model is a prevalent LOD model structure illustrated in Sec. 3.1. We use the re-aligned data Real-Data to train this model. The training details are provided in the Sec. I.2 of the Appendix. In this section, we focus on evaluating our model in the LOD scenario. We illustrate benchmark datasets, ablation studies, evaluations of existing methods, and computational cost analysis.

**Standard benchmarks.** The benchmarks we use for evaluation are OmniLabel (Schulter et al., 2023), DOD (Xie et al., 2023), RefCOCO/g/+ (*i.e.,* RefCOCO, RefCOCOg, RefCOCO+) (Yu et al., 2016; Mao et al., 2016) and OVDEval (Yao et al., 2024). OmniLabel is collected from three object detection datasets, *i.e.,* Objects365 (Shao et al., 2019), OpenImage (Kuznetsova et al., 2020), and COCO (Lin et al., 2014). It is divided into these three subsets for evaluation. There are $12.2k$ images, $20.4k$ object bbxs, and $15.8k$ expressions. The evaluation metrics are AP, AP-des-pos, and AP-des-S/M/L, which measure the average precision of object descriptions from overall, only positive, and various length perspectives. DOD contains $1k$ images with $18k$ bbxs and $422$ descriptions. It uses 'Presence' and 'Absence' to evaluate detection performance upon positive and negative queries. The RefCOCO/g/+ are from the COCO datasets with $9.9k$ images, $22.9k$ bbxs, and $46.5k$ descriptions. The details of OVDEval are shown in Sec. E. For all the benchmarks, we follow standard protocols to ensure a fair comparison.

**Real-Data.** Our Real-LOD naturally constructs a dataset via re-aligned language expressions. We randomly select images from Objects365, OpenImage, and LVIS datasets with all categories covered. There are $188k$ images with $1,346.1k$ object-query pairs in total. Among them, $473.8k$ pairs are filtered out by SigLIP, with $307.1k$ being re-aligned. The final pairs for our Real-Model training are $1,179.4k$. We name our dataset Real-Data.

### 4.1    COMPARISONS WITH STATE-OF-THE-ART LOD METHODS

We evaluate our Real-Model with existing LOD methods on the standard benchmarks, including OmniLabel, DOD, and RefCOCO/g/+ in Table 1-3. In each table, we list the vision backbones leveraged by LOD methods, source of training images (*i.e.,* 'Source'), and the number of images used for training (*i.e.,* '#Img'). We use VG, OI, O365, RefC/g/+, and CC to denote Visual Genome (Krishna et al., 2017), OpenImage (Kuznetsova et al., 2020), Objects365 (Shao et al., 2019), RefCOCO/g/+ (Yu et al., 2016), and Conceptual Captions (Sharma et al., 2018; Changpinyo et al., 2021; Xu et al., 2023), respectively. Besides, the detailed training image sources for each method can be found in Tab. 15.

In the OmniLabel benchmark, our Real-Model significantly outperforms existing LOD methods on all test sets. Especially on the OI set, under the AP-des metric, Real-Model surpasses the second-best mm-GDINO by a large margin (*i.e.,* 40.5% v.s. 23.2%). Meanwhile, under the AP-des-pos metric, Real-Model surpasses the same second-best mm-GDINO significantly (*i.e.,* 51.4% v.s. 34.5%). The superior performance of Real-Model is due to the high-quality language-object paired data provided by Real-Data. On the other hand, we observe that the training data size used for GLIP is larger than Real-Model, but the accuracy is around 50% of ours. This indicates that data quality is as important as quantity to achieve superior results.

---

[2]Suppose $N$ and $N_s$ represent the number of total expressions and correctly refined expressions, respectively. The success rate can be formulated as $\frac{N_s}{N}$.

Table 1: State-of-the-art comparisons on the OmniLabel benchmark.

| Subset | LOD method | Backbone | Source | #Img | AP-des | AP-des-pos | AP-des-S | AP-des-M | AP-des-L |
|---|---|---|---|---|---|---|---|---|---|
| COCO | MDETR (Kamath et al., 2021) | ENB3 | COCO, VG, Flickr30K | 0.3M | 13.2 | 31.6 | 15.4 | 13.5 | 12.4 |
| | GLIP (Li et al., 2022) | Swin-L | O365, OI, RefC/g/+, etc | 17.5M | 13.9 | 36.8 | 28.9 | 12.9 | 11.5 |
| | mm-GDINO (Zhao et al., 2024) | Swin-B | GoldG, O365, COCO, etc | 12M | 15.2 | 47.0 | 29.3 | 14.9 | 15.1 |
| | FIBER (Dou et al., 2022) | Swin-B | COCO, CC3M, SBU, etc | 4M | 14.3 | 38.8 | 31.3 | 12.7 | 16.1 |
| | **Real-Model** | Swin-B | Real-Data | 0.18M | **26.2** | **59.7** | **39.4** | **25.4** | **24.3** |
| O365 | MDETR (Kamath et al., 2021) | ENB3 | COCO, VG, Flickr30K | 0.3M | 3.2 | 5.9 | 3.0 | 3.2 | 2.7 |
| | GLIP (Li et al., 2022) | Swin-L | O365, OI, RefC/g/+, etc | 17.5M | 24.0 | 35.2 | 44.5 | 20.5 | 11.8 |
| | mm-GDINO (Zhao et al., 2024) | Swin-B | GoldG, O365, COCO, etc | 12M | 19.6 | 31.0 | 32.3 | 17.8 | 12.4 |
| | FIBER (Dou et al., 2022) | Swin-B | COCO, CC3M, SBU, etc | 4M | 25.9 | 38.2 | 44.7 | 22.5 | 14.1 |
| | **Real-Model** | Swin-B | Real-Data | 0.18M | **36.0** | **52.1** | **55.7** | **32.3** | **23.7** |
| OI | MDETR (Kamath et al., 2021) | ENB3 | COCO, VG, Flickr30K | 0.3M | 6.1 | 10.6 | 9.6 | 5.7 | 4.1 |
| | GLIP (Li et al., 2022) | Swin-L | O365, OI, RefC/g/+, etc | 17.5M | 20.1 | 31.2 | 33.3 | 18.7 | 10.3 |
| | mm-GDINO (Zhao et al., 2024) | Swin-B | GoldG, O365, COCO, etc | 12M | 23.2 | 34.5 | 32.3 | 23.8 | 16.9 |
| | FIBER (Dou et al., 2022) | Swin-B | COCO, CC3M, SBU, etc | 4M | 20.1 | 30.9 | 34.1 | 18.5 | 10.5 |
| | **Real-Model** | Swin-B | Real-Data | 0.18M | **40.5** | **51.4** | **54.9** | **37.8** | **30.6** |
| ALL | MDETR (Kamath et al., 2021) | ENB3 | COCO, VG, Flickr30K | 0.3M | 4.7 | 9.1 | 6.4 | 4.6 | 4.0 |
| | GLIP (Li et al., 2022) | Swin-L | O365, OI, RefC/g/+, etc | 17.5M | 21.2 | 33.2 | 37.7 | 18.9 | 10.8 |
| | mm-GDINO (Zhao et al., 2024) | Swin-B | GoldG, O365, COCO, etc | 12M | 20.8 | 33.1 | 31.9 | 19.8 | 14.1 |
| | FIBER (Dou et al., 2022) | Swin-B | COCO, CC3M, SBU, etc | 4M | 22.3 | 34.8 | 38.6 | 19.5 | 12.4 |
| | **Real-Model** | Swin-B | Real-Data | 0.18M | **36.5** | **52.1** | **54.4** | **33.2** | **25.5** |

Table 2: Evaluation results on the DOD benchmark.

| LOD method | Backbone | Source | #Img | Full | Presence | Absence |
|---|---|---|---|---|---|---|
| OWL-V2 (Minderer et al., 2023) | ViT-L | WebLI | 10B | 9.6 | 10.7 | 6.4 |
| UNINEXT (Yan et al., 2023) | ViT-H | O365, RefC/g/+ | 0.7M | 20.0 | 20.6 | 18.1 |
| GDINO (Liu et al., 2024b) | Swin-B | CC4M, O365, RefC/g/+, etc | 5.8M | 20.1 | 20.7 | 22.5 |
| mm-GDINO (Zhao et al., 2024) | Swin-B | GoldG, O365, COCO, etc | 12M | 24.2 | 23.9 | 25.9 |
| OFA-DOD (Xie et al., 2023) | RN101 | CC12M, SBU, VG, etc | 16M | 21.6 | 23.7 | 15.4 |
| APE-B (Shen et al., 2024) | ViT-L | LVIS, O365, RefC/g/+, etc | 2.6M | 30.0 | 29.9 | 30.3 |
| **Real-Model** | Swin-B | Real-Data | 0.18M | **34.1** | **34.4** | **33.2** |

Tables 2-3 shows the evaluation results on DOD and RefCOCO/g/+ benchmarks, respectively. The results are similar to those in the OmniLabel benchmark. Using a small amount of training data, our Real-Model achieves favourable results under various metrics, which surpasses existing LOD methods. This performance gain is due to our Real-Data data pairs, where diversified language expressions improve the generalizations of language and object alignment. As a result, our Real-Data datasets, with the same images and objects but diversified language descriptions, benefit Real-Model in achieving state-of-the-art performance. In addition, the evaluation results on OVDEval and the application of our method to other LOD models are presented in Sec. E.

## 4.2 ABLATION STUDY

We train our Real-Model by using three training data pair configurations (*i.e.,* A, B, and C forms) and evaluate the corresponding LOD performance on the OmniLabel benchmark. We randomly select $94k$ images from O365 and OI datasets covering all categories, which is a subset of Real-Data. These images, together with target objects and raw expressions, constitute our original training data pairs with an amount of $933k$ (*i.e.,* A form). Moreover, we use SigLIP to filter out some data pairs where expressions do not match the target object. The remaining pairs are $695k$ (*i.e.,* B form). Furthermore, we use Real-LOD to re-align mismatched pairs to add them back to B, which increases the number of pairs to $863k$ (*i.e.,* C form). We use data pairs in A, B, and C forms to train Real-Model separately and evaluate the corresponding performance. This helps analyze how our Real-LOD improves LOD from a data-alignment perspective.

Tab.4 shows the LOD results via three data configurations (*i.e.,* A, B, and C forms). It demonstrates that on the COCO test set, Real-Model achieves a $21.2\%$ AP by using all the training data pairs (*i.e.,* A form). After removing data pairs with mismatched expressions, Real-Model increases to $22.2\%$ (*i.e.,* B form). This improvement indicates that data quality essentially benefits LOD performance. Then, our Real-LOD refines filtered data pairs for a supplement, which improves Real-Model to $24.2\%$ (*i.e.,* C form). It shows that Real-LOD increases data quantity with high quality, leading to further LOD improvement. The results on the other two test sets (*i.e.,* O365 and OI) indicate

Table 3: Evaluation results on the RefCOCO/g/+ benchmark. '*' indicates that the model employs RefCOCO/g/+ for training.

| LOD method | Backbone | Source | #Img | RefCOCO | | | RefCOCO+ | | | RefCOCOg | |
|---|---|---|---|---|---|---|---|---|---|---|---|
| | | | | val | testA | testB | val | testA | testB | val-u | test-u |
| MDETR (Kamath et al., 2021) | ENB3 | COCO, VG, Flickr30K | 0.3M | 73.4 | - | - | 58.8 | - | - | 57.1 | - |
| APE-A (Shen et al., 2024) | ViT-L | COCO, LVIS, O365, etc | 2.0M | 34.2 | 34.8 | 36.1 | 33.5 | 32.3 | 36.0 | 38.9 | 40.5 |
| **Real-Model** | Swin-B | Real-Data | 0.18M | 74.0 | 79.6 | 66.0 | 76.4 | 83.1 | 68.5 | 80.8 | 81.2 |
| GLIP* (Li et al., 2022) | Swin-L | O365, OI, RefC/g/+, etc | 17.5M | 53.1 | 59.4 | 46.8 | 54.0 | 59.4 | 47.0 | 60.7 | 60.4 |
| GDINO* (Liu et al., 2024b) | Swin-B | CC4M, O365, RefC/g/+, etc | 5.8M | - | - | - | 73.6 | 82.1 | 64.1 | 78.3 | 78.1 |
| APE-B* (Shen et al., 2024) | ViT-L | LVIS, O365, RefC/g/+, etc | 2.6M | 84.6 | 89.2 | 80.9 | 76.4 | 82.4 | 66.5 | 80.0 | 80.1 |
| **Real-Model*** | Swin-B | RefC/g/+, Real-Data | 0.24M | **91.3** | **93.1** | **88.0** | **85.4** | **90.3** | **78.6** | **88.4** | **89.0** |

Table 4: Ablation study on OmniLabel benchmark. Our training data pairs consist of images, target objects, and expressions (expr). We adjust training data pairs by processing raw expr differently (*i.e.,* SigLIP filter and Real-LOD) and evaluate the corresponding performance. Note that we use a subset of Real-Data.

| Test subset | Training data type | #Img | AP-des | AP-des-pos | AP-des-S | AP-des-M | AP-des-L |
|---|---|---|---|---|---|---|---|
| | raw expr (A) | $933k$ | 21.2 | 59.4 | 31.3 | 21.1 | 18.6 |
| COCO | raw expr w.filter (B) | $695k$ | 22.2 | 59.4 | 32.4 | 21.9 | 19.4 |
| | raw expr w.filter + Real-LOD (C) | $863k$ | **24.2** | **59.6** | **35.2** | **24.2** | **21.1** |
| | raw expr (A) | $933k$ | 27.6 | 43.1 | 39.8 | 25.5 | 17.9 |
| O365 | raw expr w.filter (B) | $695k$ | 28.5 | 43.7 | 40.9 | 26.2 | 18.5 |
| | raw expr w.filter + Real-LOD (C) | $863k$ | **32.4** | **48.5** | **47.5** | **30.0** | **21.3** |
| | raw expr (A) | $933k$ | 30.5 | 43.0 | 37.2 | 30.3 | 23.2 |
| OI | raw expr w.filter (B) | $695k$ | 31.4 | 43.5 | 38.1 | 31.2 | 24.0 |
| | raw expr w.filter + Real-LOD (C) | $863k$ | **33.5** | **44.9** | **42.2** | **32.9** | **24.8** |

similar phenomena. When training Real-Model, data quality also has an influential impact on LOD performance, especially when the data quantity is increasing. Our Real-LOD re-aligns mismatched object and language pairs to increase data quantity while preserving data quality. To this end, our Real-Model learned with re-aligned data in C form performs best on the OmniLabel benchmark.

### 4.3 ANALYSIS ON COMPUTATIONAL COST

In our Real-LOD, we also leverage two strategies to further mitigate workflow time costs: 1) We leverage SigLIP to exclude 75% of training data from our workflow, leaving only 25% to be processed. 2) We set the max cycle number of our workflow as 4 to trade off time cost and performance. We elaborate on our computation cost in Tab. 13. For refining one expression, we report the average number of calls for each step and the time cost during each call. The time cost is reported based on 48 V100 32G GPUs for our workflow execution. For refining one expression, our workflow takes 1.579 seconds in total, with an average cycle number of 3.08. We also provide the distribution of iteration numbers in Fig. 18. Note that the max iteration number here is 10 for investigation. In addition, our workflow is completely offline without bringing additional computational burden to LOD model inference.

## 5 CONCLUSIONS

Re-aligning language to visual objects has been developed from manual descriptions to automatic VLM generations. The data pairs are scaling up to advance the connection performance of LOD. The generated descriptions may not match the objects due to model hallucinations. We thus propose Real-LOD to refine the language expressions gradually via agentic workflows. The data quality is preserved along with the increased data quantity. We train a prevalent LOD model using our data to largely surpass existing LOD methods. Our automatic workflow contains the expanding potential to re-align language descriptions of any objects. With an open vocabulary detector to locate objects with short category labels and VLMs to expand expression, our Real-LOD will continuously produce high-quality training pairs to scale up LOD performance.

## ACKNOWLEDGMENT

This work was funded by NSFC (No. 62225604, 62176130), the Science and Technology Support Program of Tianjin, China (No. 23JCZDJC01050). The Supercomputing Center of Nankai University partially supported computation.

## ETHICAL STATEMENT

We declare that our research does not present any potential ethical issues. The study does not involve human subjects, sensitive data, or methodologies that could result in harmful outcomes or biases. All data used in this work is publicly available, and no privacy or security concerns are implicated.

## REPRODUCIBILITY STATEMENT

Transparency and reliability are crucial to our research. In this statement, we summarize the measures taken to facilitate the reproducibility of our work and provide references to the relevant contents in the main paper and appendix.

**Source code.** We intend to make our source code, model weights, and datasets available to the public following the paper's acceptance. It will allow the following researchers to access and utilize our code to reproduce our experiments and results. The detailed installation and execution instructions will be listed in 'README.md'.

**Experimental setup.** We provide the basic implementation information of our Real-LOD in Sec. 3.1 and Sec. 3.2. Besides, we provide the experimental setup and evaluation settings in Sec. 4 and Sec. I.3. The details of Real-Data are listed in the Sec. 3.3 and Sec. 4. Moreover, the training and architectural details of Real-Model can be found in the Sec. I.2 and Sec. I.4 of the Appendix.

We provide the above resources and references to ensure the reproducibility of our work. It enables fellow researchers to verify our method. We also welcome any inquiries or requests for further clarification on our methods.

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

## APPENDIX OVERVIEW

We provide an overview to present a clear understanding of this section.

- In Sec. A, we provide an overview of the pipeline for language data generation.
- In Sec. B, we show more examples of raw expressions corrected by our Real-LOD.
- In Sec. C, we present visual comparisons of existing LOD methods under various queries.
- In Sec. D, we illustrate several examples of how Real-LOD refines raw expressions.
- In Sec. E, we provide additional evaluation results on the LOD benchmark.
- In Sec. F, we present a pseudo-code of proposed Real-LOD workflow.
- In Sec. G, we show prompts for LLM and VLM to execute different tasks.
- In Sec. H, we provide the statistical results of the computation cost.
- In Sec. I, we outline the specifics of the training, evaluation, datasets, and model structure.
- In Sec. J, we provide additional discussion of our paper.
- In Sec. K, we provide more analytical experiment for Real-LOD.

## A    EXPRESSION GENERATION PIPELINE

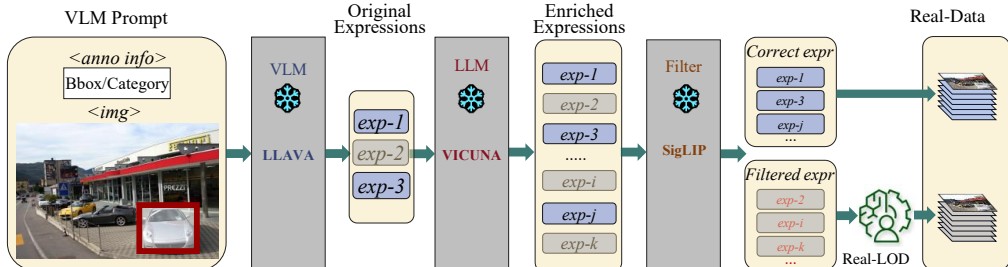

Figure 9: An overview of our language generation pipeline for Real-Data. In this pipeline, we first use LLaVA-v1.6-34B (Liu et al., 2024a) to generate descriptions. For each object, we randomly select two prepared prompts presented in Tab. 8 with an image and corresponding category for LLaVA to generate expressions. Second, Vicuna-v1.5-13B (Zheng et al., 2023) is introduced to generate synonyms to expand the number of expressions using the prompt in Tab. 8. We repeat the process two times for each expression. Then, we use SigLIP to filter expression-image pairs with low scores. Finally, we maintain correct data pairs and refine filtered expressions via our Real-LOD to build the final dataset Real-Data.

# B    RE-ALIGNMENT EXAMPLES OF REAL-LOD

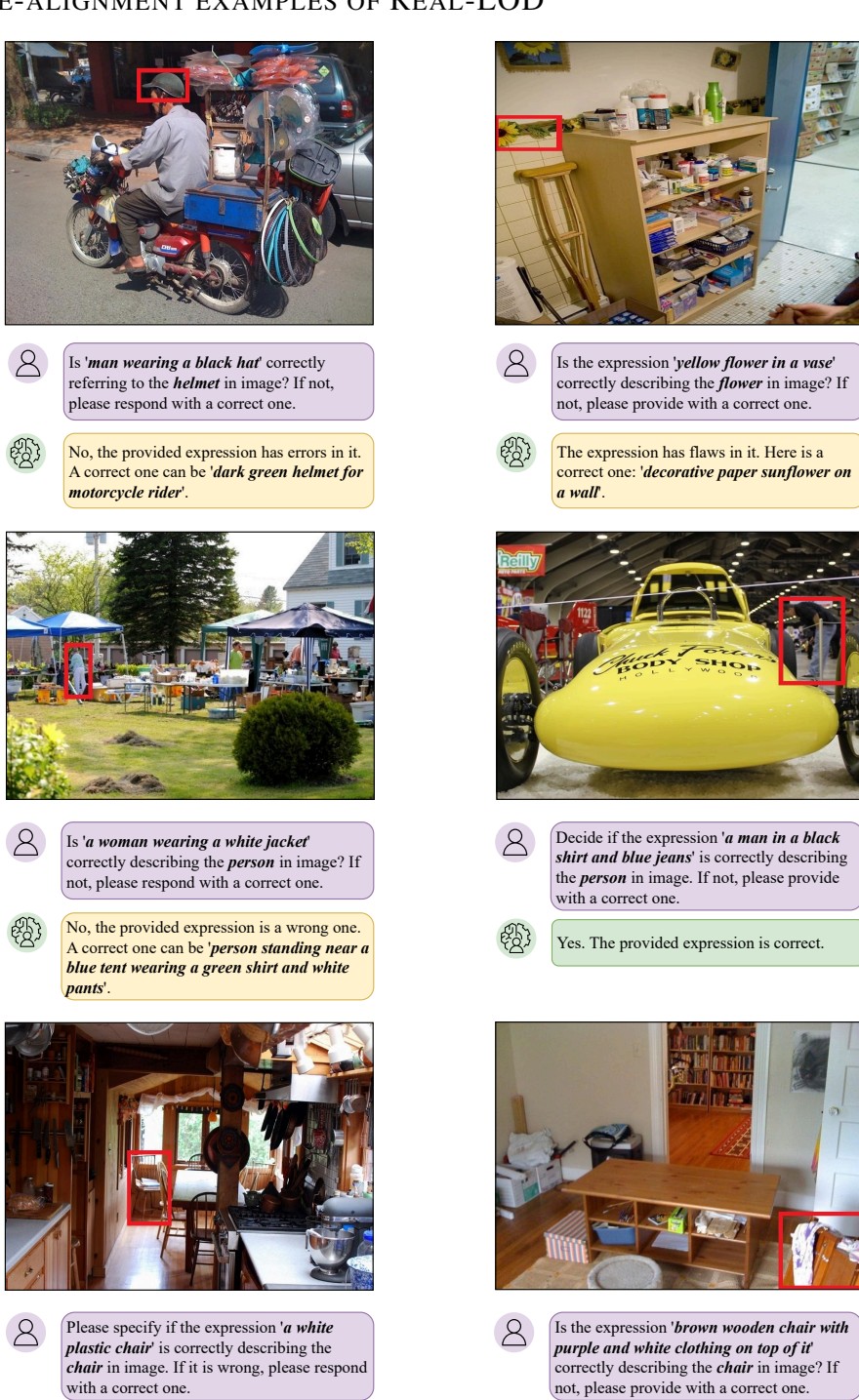

Figure 10: We show examples of the re-alignment by Real-LOD. Real-LOD can correct wrong expressions and remain correct ones.

## C  VISUAL COMPARISON RESULTS OF LOD MODELS

**Query:** "This item is used to keep warm in colder weather."

**Query :** "Woman in wedding dress next to a man in suit."

**Query :** "Pillow placed at the head of the bed."

**Query :** "These two people each have a pink surfboard."

**Query :** "The fire extinguisher on the left."

**Query :** "Cows that are laid down."

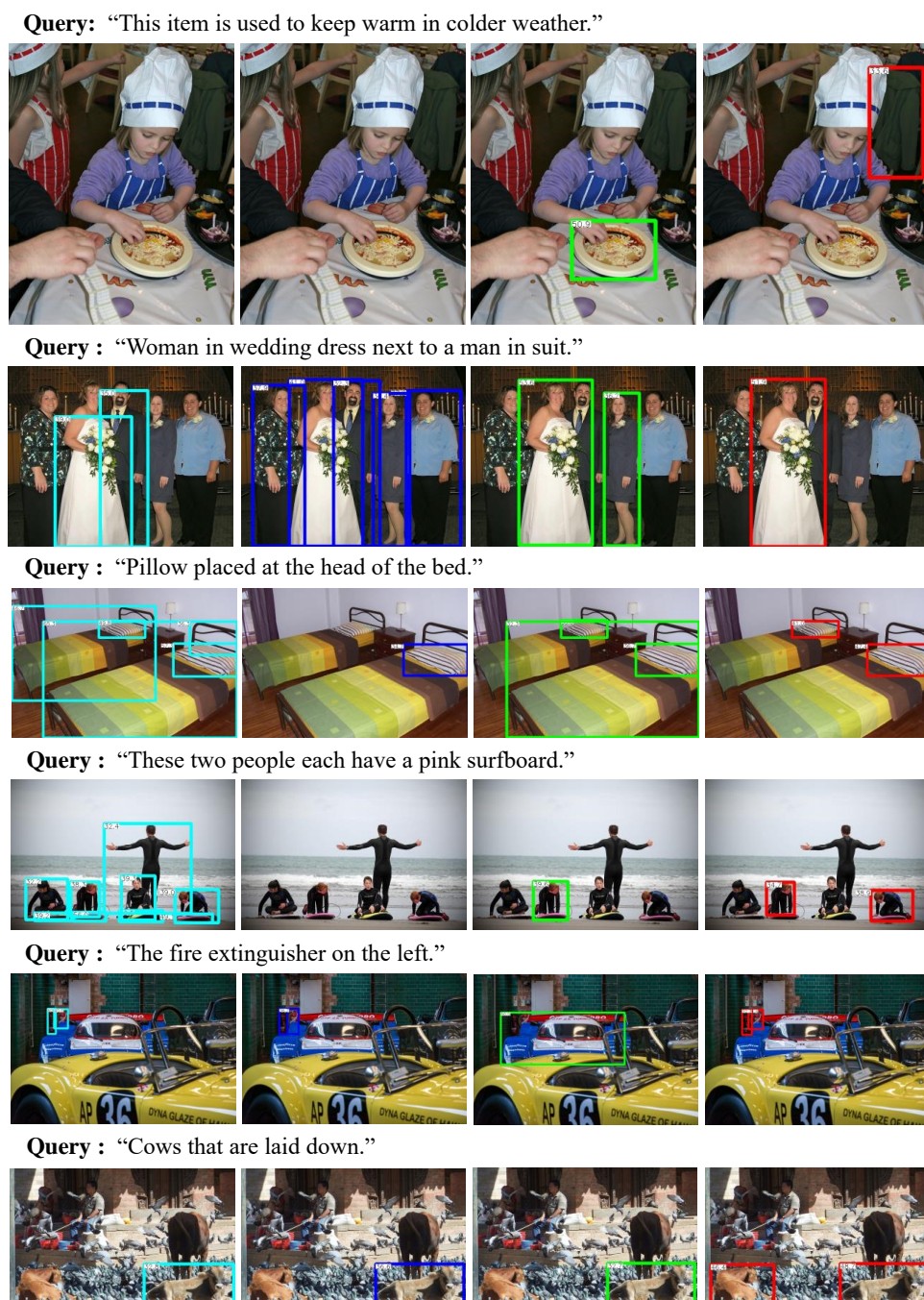

   (a) GLIP-L      (b) APE-B     (c) mm-GDINO    (d) **Real-Model**

Figure 11: Visual comparison with existing language and vision detectors. The backbone of GLIP and APE-B is ViT-L, and the backbone of mm-GDINO is Swin-B. We use 0.3 as the score threshold for the fair comparison. For convenience, we use bbxs with different colors to distinguish each model. The color we used for Real-Model is red.

## D    EXAMPLES OF RE-ALIGNMENT BY REAL-LOD

In Fig. 12- 17, we show several examples of how Real-LOD works.

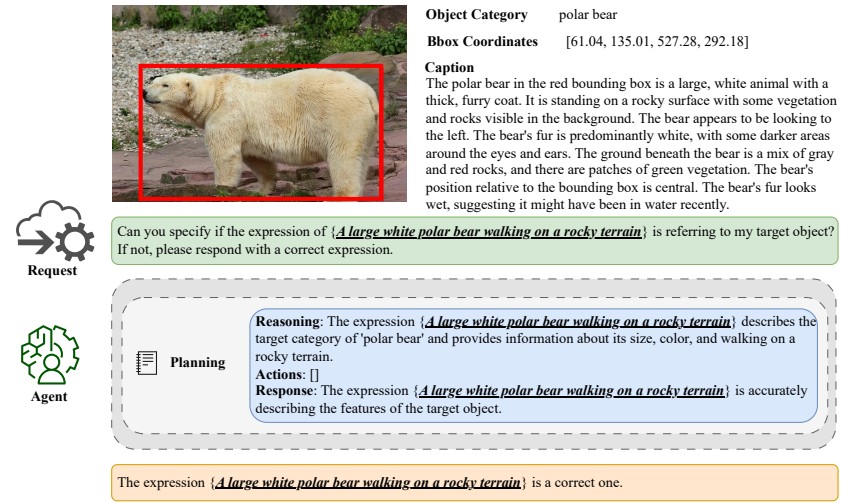

Figure 12: An example of how Real-LOD handles correct expressions.

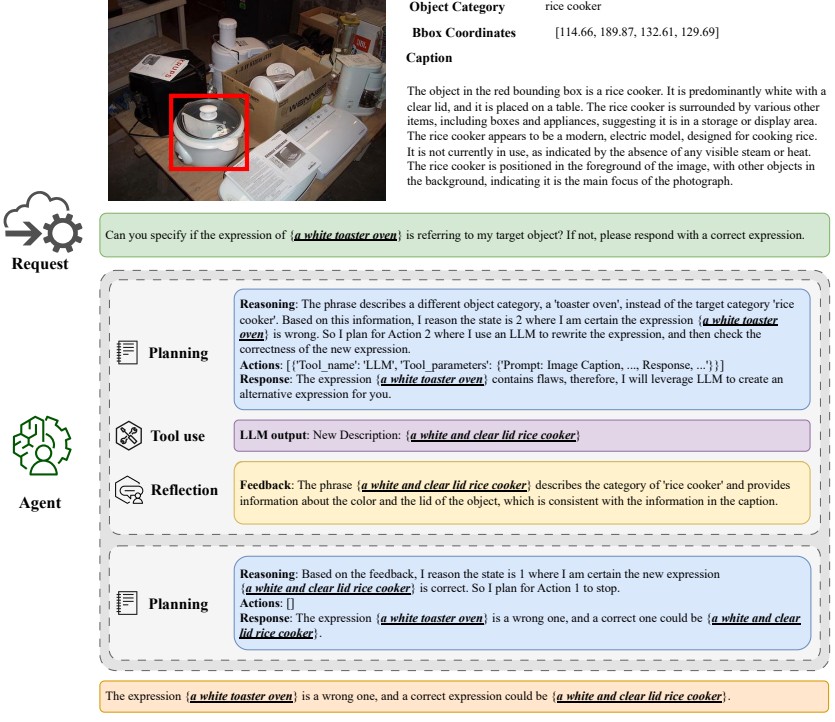

Figure 13: An example of how Real-LOD corrects a wrong expression. It consists of the 'Rewrite' action.

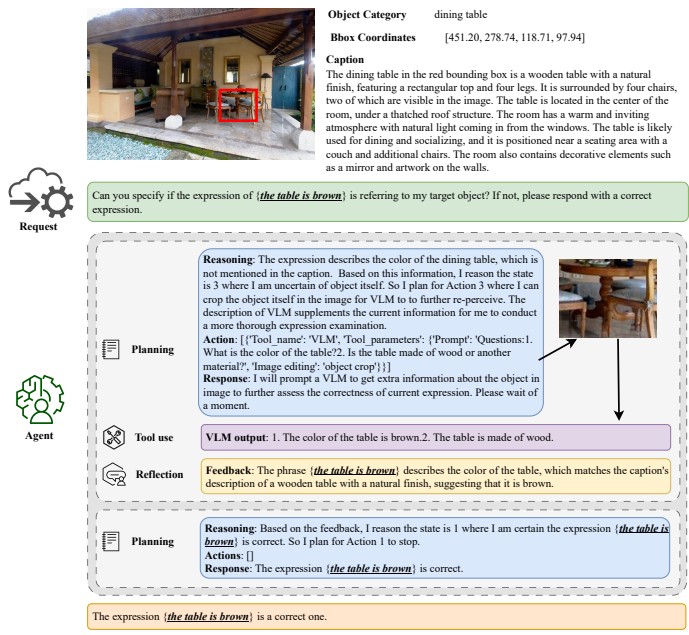

Figure 14: An example of how Real-LOD handles an uncertain expression. It consists of the 'VLM with object crop' action.

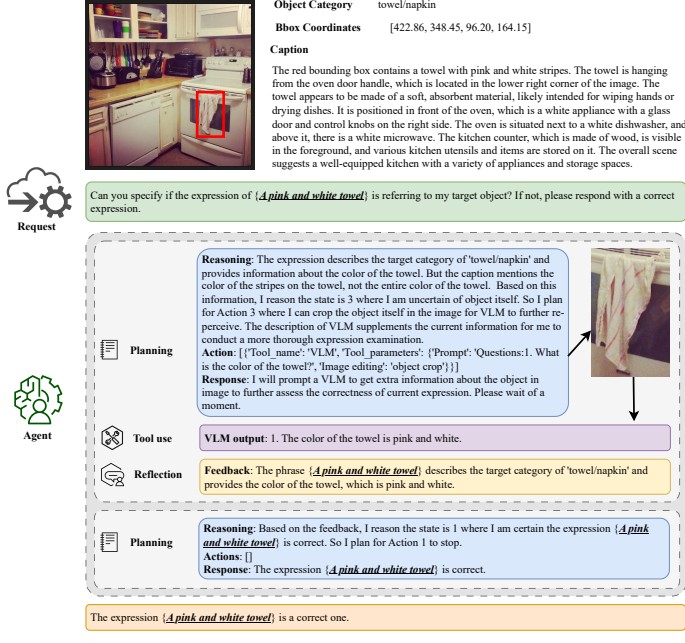

Figure 15: An example of how Real-LOD handles an uncertain expression. It consists of the 'VLM with object crop' action.

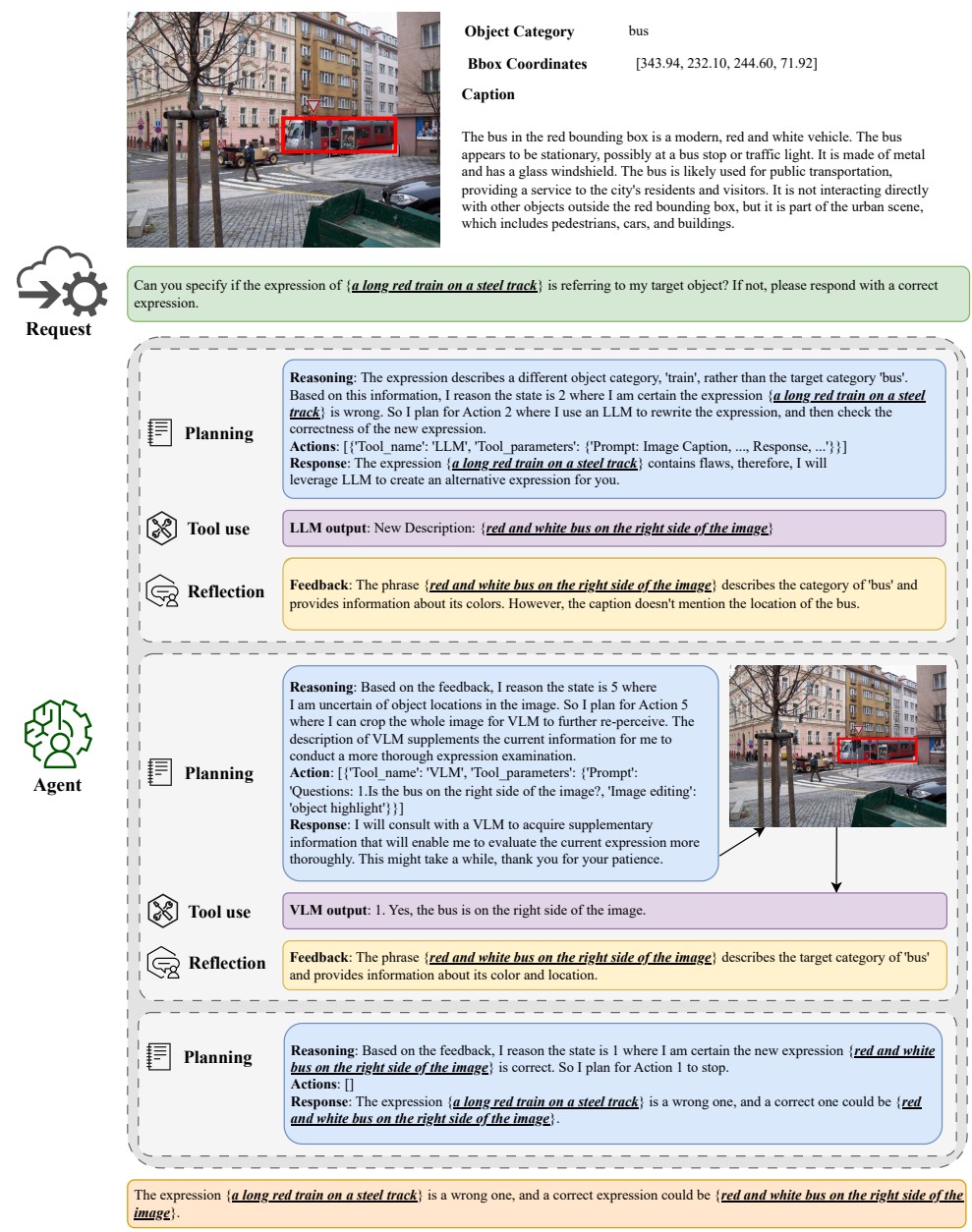

Figure 16: An example of how Real-LOD refines a raw expression. This workflow consists of two actions ('Rewrite' and 'VLM with object highlight')

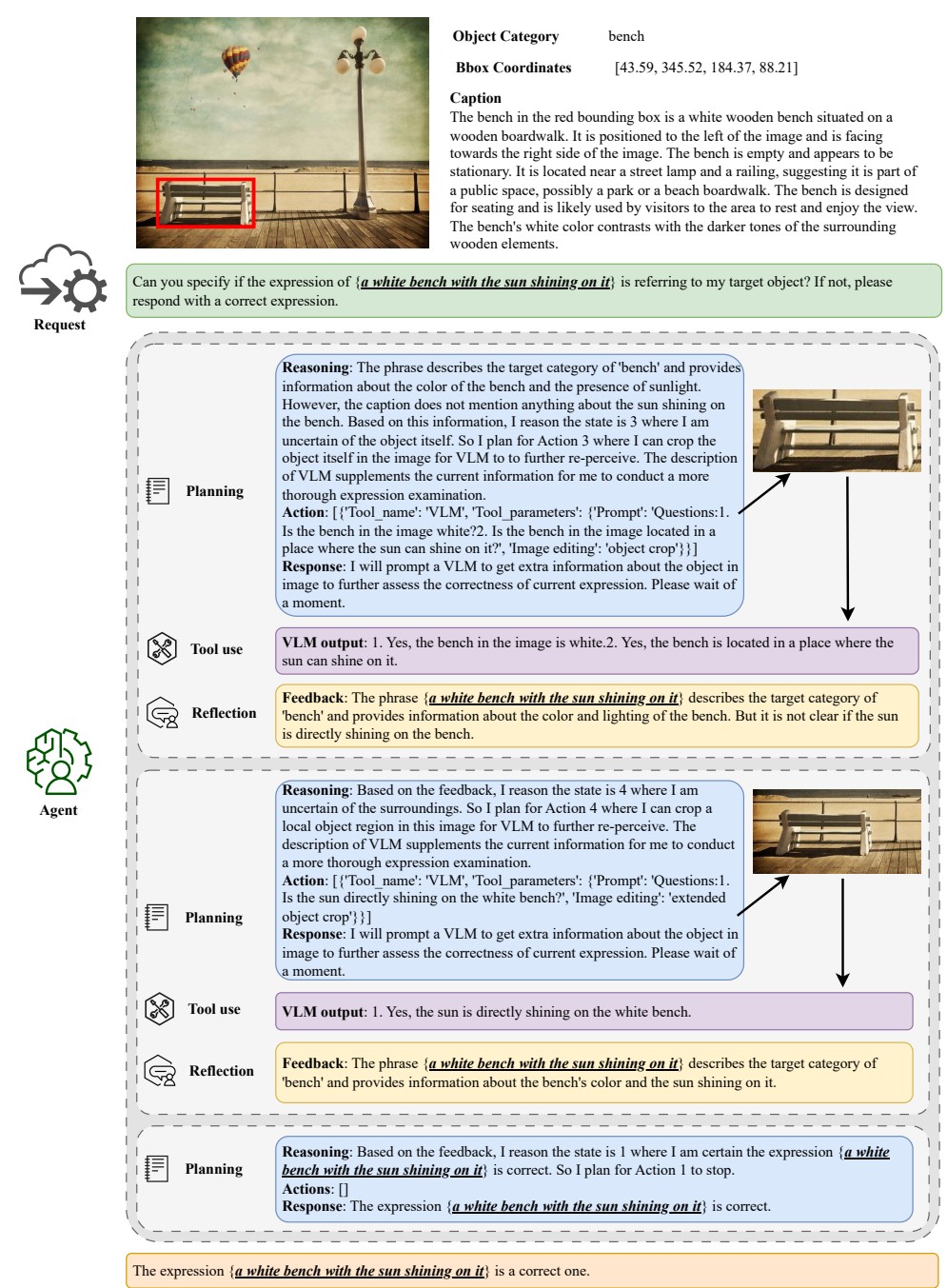

Figure 17: An example of how Real-LOD handles an uncertain expression. It consists of 'VLM with object crop' and 'VLM with extended object crop' actions.

# E ADDITIONAL EVALUATION RESULTS

## E.1 APPLICATION TO OTHER LOD MODELS

In this subsection, we conduct experiments to demonstrate the generalization ability of our method. We apply our Real-Data to UNINEXT (Yan et al., 2023) and a tiny version of mm-GDINO (Zhao et al., 2024). We report the results of OmniLabel and DOD benchmarks in Tab. 5 and Tab. 6, respectively. For UNINEXT, the AP-des of OmniLabel and of DOD are significantly increased to 23.4% and 24.2% without adjusting any training parameters. Notably, we only train UNINEXT for five epochs based on a relatively small backbone (*i.e.,* ResNet-50). For mm-GDINO with a Swin-T backbone, the AP-des of OmniLabel and of DOD are significantly improved to 29.9% and 30.8%. The results demonstrate the holistic nature of our method.

Table 5: Application to other LOD models on OmniLabel benchmark.

| Test subset | LOD method | BackBone | Real-Data | AP-des | AP-des-pos | AP-des-S | AP-des-M | AP-des-L |
|---|---|---|---|---|---|---|---|---|
| COCO | MM-GDINO (Zhao et al., 2024) | Swin-T | | 10.1 | 31.2 | 19.7 | 9.4 | 10.4 |
| | | | ✓ | **20.5** | **51.9** | **30.9** | **19.6** | **21.1** |
| | UNINEXT (Yan et al., 2023) | ResNet-50 | | 3.6 | 10.3 | 8.9 | 3.8 | 2.0 |
| | | | ✓ | **14.6** | **41.4** | **24.8** | **13.6** | **14.8** |
| O365 | MM-GDINO (Zhao et al., 2024) | Swin-T | | 16.1 | 24.7 | 32.7 | 12.7 | 7.9 |
| | | | ✓ | **29.6** | **44.4** | **49.0** | **26.2** | **19.6** |
| | UNINEXT (Yan et al., 2023) | ResNet-50 | | 7.6 | 13.9 | 10.8 | 8.1 | 4.6 |
| | | | ✓ | **24.3** | **38.2** | **37.7** | **21.3** | **17.3** |
| OI | MM-GDINO (Zhao et al., 2024) | Swin-T | | 20.6 | 30.4 | 37.2 | 18.1 | 12.9 |
| | | | ✓ | **33.7** | **45.1** | **48.5** | **30.8** | **25.1** |
| | UNINEXT (Yan et al., 2023) | ResNet-50 | | 5.1 | 7.1 | 9.2 | 4.9 | 3.2 |
| | | | ✓ | **25.3** | **36.3** | **36.9** | **22.8** | **19.1** |
| ALL | MM-GDINO (Zhao et al., 2024) | Swin-T | | 17.0 | 26.7 | 33.3 | 13.9 | 9.3 |
| | | | ✓ | **29.9** | **44.3** | **47.5** | **26.9** | **20.7** |
| | UNINEXT (Yan et al., 2023) | ResNet-50 | | 6.0 | 10.4 | 9.6 | 5.8 | 3.5 |
| | | | ✓ | **23.4** | **37.3** | **36.3** | **20.7** | **17.3** |

Table 6: Application to other LOD models on DOD benchmark.

| LOD method | BackBone | Real-Data | Full | Presence | Absence |
|---|---|---|---|---|---|
| MM-GDINO (Zhao et al., 2024) | Swin-T | | 23.0 | 21.9 | 26.0 |
| | | ✓ | **30.8** | **30.3** | **32.7** |
| UNINEXT (Yan et al., 2023) | ResNet-50 | | 10.7 | 10.6 | 10.9 |
| | | ✓ | **24.2** | **23.2** | **26.9** |

## E.2 COMPARISONS WITH STATE-OF-THE-ART LOD MODELS ON OVDEVAL BENCHMARK

In this subsection, we show evaluation results in OVDEval benchmark (Yao et al., 2024). The benchmark contains $15.1k$ images with $28.1k$ bbxs and $10.1k$ language expressions. The dataset is divided into several sub-datasets according to aspects such as 'color' and 'relationship'. We select a language-based sub-dataset to compare our Real-Model model with other detectors. Tab. 7 shows the results. The OmDet method performs best in the 'Relationship' sub-dataset. The reason is that the testing data is collected from HOI (Chen et al., 2023) dataset, which is used to train the OmDet model. Our Real-Model outperforms existing LOD models on average in this benchmark.

Table 7: State-of-the-art comparison on OVDEval benchmark. We report evaluation results AP (%) of each sub-dataset. 'Source' refers to the source of training images. '#Img' refers to the number of images.

| LOD method | Backbone | Source | #Img | color | material | Position | Relationship | Negation | Avg |
|---|---|---|---|---|---|---|---|---|---|
| GLIP (Li et al., 2022) | Swin-L | O365, OI, RefC/g/+, etc | 17.5M | 6.7 | 15.8 | 48.1 | 33.2 | 51.8 | 31.1 |
| OmDet (Yao et al., 2024) | ConvNext-B | O365, GoldG, HOI-A, etc | 1.1M | 24.5 | 22.5 | 47.7 | **51.8** | 55.8 | 40.4 |
| FIBER (Dou et al., 2022) | Swin-B | COCO, CC3M, SBU, etc | 4M | 9.4 | 17.7 | 48.1 | 33.2 | 58.1 | 33.3 |
| **Real-Model** | Swin-B | Real-Data | 0.18M | **25.7** | **22.5** | **59.3** | 41.9 | **68.4** | **43.6** |

# F ALGORITHM OF AGENTIC WORKFLOW

**Algorithm 1** Pseudo code of Real-LOD. We show the detailed code of our workflow in flexibly leveraging tools to re-align raw expressions to given objects.

**Input:** image $\mathbf{I}$, object locations $\mathbf{O}$, caption $\mathbf{C}$, raw expression $\mathbf{E_r}$
**Output:** re-aligned expression $\mathbf{E_R}$
1: $agent \leftarrow$ Real-LOD(init $assistant$, init $vlm\_tool$, init $llm\_tool$) //Agent initialization
2: $info\_pool \leftarrow \{$"image": $\mathbf{I}$, ..., "expressions": $[\mathbf{E_r}]\}$ //Initialized as input
3: $iteration \leftarrow 0$
4: $stop \leftarrow$ False
5: $solved \leftarrow$ False
6: **while** not $stop$ **do**
7:     //Stage 1. planning based on current $info\_pool$
8:     $reasoning, actions, values \leftarrow agent.assistant$
9:     //Stage 2. tool use and update $info\_pool$
10:     **if** ($actions$ is not empty) and ($iteration < max\_iter$) **then**
11:         **for** $action$ in $actions$ **do**
12:             $tool\_name, tool\_params \leftarrow action$
13:             **case** $tool\_name$ is "VLM", update $info\_pool$ with $agent.vlm\_tool(tool\_params)$
14:             **case** $tool\_name$ is "LLM", update $info\_pool$ with $agent.llm\_tool(tool\_params)$
15:         **end for**
16:         //Stage 3. reflection on tool outputs
17:         update $feedback$ with $agent.llm\_tool$
18:         $iteration \leftarrow iteration + 1$
19:     **else if** $actions$ is empty **then**
20:         $solved \leftarrow$ True // reach a correct expression
21:     **end if**
22:     $stop \leftarrow solved$ or ($iteration == max\_iter$)
23: **end while**
24: $\mathbf{E_R} \leftarrow info\_pool["expressions"][-1]$
25: **return** $\mathbf{E_R}$

# G PROMPTS FOR LLM AND VLM

## G.1 PROMPTS FOR RAW EXPRESSION GENERATION

**Prompts for VLM to generate raw expression**
1. For the given image <image>, please provide a unique description for the <object> in the area <boxes>.
2. What is the content depicted of the <object> located in the area <boxes> of the image <image>?
3. Please describe the <object> in the area <boxes> of this image <image>.
4. I would like to know the description of the <object> in the area <boxes> of the picture <image>.
5. Kindly describe the <object> in the area <boxes> of the picture <image>.
6. Give me detailed descriptions of the <object> in area <boxes> of this image <image>, including its color, material, attribute, etc.
7. What's the difference between <object> in area <boxes> and other <object> in this image <image>?
8. What is the relative position of the <object> in area <boxes> of the picture <image>?
9. What's the relationship of <object> in area <boxes> with its' surrounding objects?

**Prompt for LLM to diversify raw expression**
I want you to act as a synonymous expression provider. I will give you a text of phrase or short sentence, which is an expression that describes a main object while mentioning some other objects. And you will reply to me with a new expression that has the same semantic meaning and describes the same main object as the provided expression. The new expressions should also be phrases or short sentences no longer than 25 words. Do not write explanations on replies. Do not repeat.

Table 8: Prompts for raw expression generation.

## G.2 Prompts for rewrite task

---

**Task Description**
You are an excellent text analyst and generator. I want a short text description that correctly describes my chosen object in an image. Now I already have one description, but there might be mistakes in it, and I want you to help with this. I will provide you with the following as background knowledge:
1. Image Caption: a caption describing the content in an image.
2. Chosen Object: one or more objects chosen in this image to describe and their corresponding coordinates. The top-left corner of the image has coordinates [0, 0] and the bottom-right corner has coordinates [1, 1]. Each object is represented as {"id": unique object identification, 'category name': object category, 'box':[top-left x, top-left y, box width, box height]}.
3. Other Object: other objects in image and their corresponding coordinates, which are provided to you in the same format as the 'Chosen Object'.
4. Object Description: a short text for you to analyze (this description could be correct, partially correct or wrong).

As an assistant, analyze the 'Object Description' and generate a 'New Description' based on it, which correctly describes the chosen object:
1. Your new description should be centered on the chosen object, and describe the correct object category
2. Your new description should be consistent with information provided by the given caption and with general knowledge
3. Your new description should be a short phrase and has reasonable meaning

**One In-context Example**
Prompt: Image Caption: The image showcases a clean and well-organized kitchen with wooden floors and wooden cabinets. The kitchen features a white refrigerator, a sink with a window above it, and a stove with an oven below it. Additionally, there is a dishwasher and a washing machine in the kitchen. The appliances are neatly arranged, and the overall appearance of the kitchen is tidy and inviting.
Chosen Object: [{'id':0, 'category name':'Home appliance', 'box':[0.68, 0.52, 0.15, 0.05]}, {'id':1, 'category name':'Home appliance', 'box':[0.0, 0.26, 0.24, 0.74]}, {'id':2, 'category name':'Home appliance', 'box':[0.53, 0.56, 0.14, 0.28]}, {'id':3, 'category name':'Home appliance', 'box':[0.68, 0.22, 0.15, 0.06]}, {'id':4, 'category name':'Home appliance', 'box':[0.67, 0.57, 0.16, 0.21]}]
Other Object: []
Object Description: 'appliance for cleaning dishes with water and detergent'
 Response: Reasoning: The description is centered on home appliances, but it only refers to the washing machine in the kitchen, which is only one of the chosen objects. The object category is 'home appliance', which also includes the refrigerator, the sink and the stove mentioned in the given caption. So the description does not match all the objects that belong to 'home appliance' in the image. I can rewrite this description into a more specific one.
New Description: 'home appliances of refrigerator, sink, stove, dishwasher and washing machine'

**Python code together with above text prompts are directly sent to LLM-for-Rewriting**

```python
messages = [{"role": system, "content":task_description}]
for example in in_context_examples:
    messages.append({"role":user, "content":example["prompt"]})
    messages.append({"role":assistant, "content":example["response"]})
content=f"Image caption: " + caption + "\n"
content+=f"Chosen Object: " + chosen_object + "\n"
content+=f"Other Object: " + other_object + "\n"
content+=f"Object Description: " + object_description + "\n"
messages.append({"role": user, "content": content})
```

Table 9: Task description and an in-context example for LLM-for-Rewriting tool. There are five in-context examples used in the text prompt, we show one example of them here for simplicity.

## G.3 PROMPTS FOR REFLECTION

**Task Description**
You are an excellent text analyst. I want to get correct descriptions of a target object in an image. Now I already have a phrase describing this target object, but the texts might contain mistakes and I want you to help with this.
I will provide you with this phrase to be checked, along with an 'Object Category' and a 'Caption' as reference information:
1. Object Category: the exact category name of my target object.
2. Caption: a long text describing content in the image, information provided in this caption is correct. Note that some details in the image might be missing in this caption.
Given the reference information, your task is to verify if the phrase correctly describes my target object.

## Process Instruction
1. Correct phrase: If the phrase describes the target category and provides similar information that can be found in the caption, this phrase is correct.
2. Uncertain phrase: If the phrase describes the target category but provides information that is missing in the caption, this phrase is uncertain. Please tell me what information in this phrase is not mentioned in the given caption. Extra information could be object color, object material, object location in the image, object action, object relation with other objects in the image, etc.
3. Wrong phrase: If the phrase describes a different object category, or the phrase provides information that conflicts with the caption, this phrase is wrong.

**One In-context Example**
Prompt: Object Category: Laptop
Caption: In the image, there are two women sitting at a table, both focused on their laptops. One of the women is holding up her middle finger, possibly as a gesture of defiance or humor. On the table, there is a bottle, a cup, and a laptop being used by one of the women. The woman with the laptop is wearing a scarf, and the other woman is positioned on the other side of the table. Drink: A clear plastic bottle of water is on the table. It is placed in front of the woman with the laptop. Laptop: The laptop is a black and silver Dell computer. The woman is using it while sitting at the table. Table: The table is a dining table where the woman is sitting with her laptop and a bottle of water. Bottle: The bottle is a clear plastic bottle of water. It is placed on the table in front of the woman with the laptop.
Phrase: 'black and silver Dell computer'
Response: Feedback: The phrase describes a computer and provides its color and brand. 'Computer' is similar with 'Laptop' and the caption claims the laptop in the image to be a black and silver Dell computer, this phrase is correctly describing the target object.

**Python code together with above text prompts are directly sent to LLM-Reflector**

```python
messages = [{"role": system, "content":task_description}]
for example in in_context_examples:
    messages.append({"role":user, "content":example["prompt"]})
    messages.append({"role":assistant, "content":example["response"]})
content=f"Object Category: " + category + "\n"
content+=f"Caption: " + caption + "\n"
content+=f"Phrase: " + phrase + "\n"
messages.append({"role": user, "content": content})
```

Table 10: Task description and an in-context example for agent reflection. There are five in-context examples used in the text prompt. We show one example of them here for simplicity.

### G.4 Prompts for Response Generation of Fine-tuning Data

---

**1. Generate question for uncertain state**

**Task Description**
You are an excellent text analyst.
I have a phrase that describes an object in an image. I have already analyzed this phrase based on a caption. And I have reached the conclusion that this phrase is uncertain. I want you to help me generate questions asking about the missing information in the caption.
I will provide you with my statement of the reason why the phrase is uncertain. As an assistant, please analyze my statement and respond with your generated questions. Ask about the object in the image only. Do not involve the caption in your questions. Ask as few questions as possible.

**One In-context Example**
Prompt: Phrase: "a book with the word 'Fix' on it"
Reasoning: The phrase "a book with the word 'Fix' on it" describes the target category of 'book' and provides extra information that the word "Fix" is on the book. However, the caption does not mention any specific word on the book, so extra information is needed to determine if the phrase is correct.
Conclusion: The phrase is uncertain. The caption does not mention any specific word on the book, so it is unclear if the phrase is correct or not.
 Response: Questions: 1. Is there any specific word on the book? If so, what is the word on the book?

**2. Generate target object for action during uncertain state**

**Task Description**
You are an excellent text analyst. I have a phrase that describes an object in an image. I have already analyzed this phrase based on a caption. And I have reached the conclusion that this phrase is uncertain.
First, I want you to help me generate questions asking about the missing information in the caption. I will provide you with my statement of the reason why the phrase is uncertain. As an assistant, please analyze my statement and respond with your generated questions. Ask about the object in the image only. Do not involve the caption in your questions. Ask as few questions as possible.
Secondly, based on the questions you have generated, I want you to give me a target area where the missing information may most likely be found. Choose one receptive field from the following 3 options: 1) the object itself, 2)the object and the surrounding areas, 3)the whole image. Here are some hints for you. If the missing information is about some attribute of the object, such as colors, materials, the target area is likely the object itself 1). If the missing information is something interacting with the object, the target area is likely the object and the surrounding area 2). If the missing information is about another object, the target area is likely the whole image 3). Give me the number of the choices, and only choose one target area per task.

**One In-context Example**
Prompt: Phrase: 'bread loaf on a tan bowl'
Reasoning: The phrase describes a large tan colored bowl with a bread loaf sitting in it. However, the caption does not mention the color or the presence of food in the bowl. The size of the bowl is also not mentioned.
Conclusion: The phrase is uncertain. The color of the bowl, the presence of food and the size of the bowl are not mentioned in the caption.
Questions: 1. What is the color of the bowl? 2. Is there any food in the bowl? If so, what type of food is in the bowl? 3. What is the size of the bowl? Is it small or large?
 Response: Target area: 1)the object itself.

---

Table 11: Task description and an in-context example for response generation of fine-tuning data. There are five in-context examples used in each text prompt. We show one example here for simplicity. We omit the Python code together with text prompts, which is similar to Tab. 9 and Tab. 10.

## G.5 VISUAL AND LANGUAGE PROMPTS FOR VLM

> **Language prompt**
> Here are some prior knowledge about this image. The object in the image is a {*object category*}. There is a possible description of this image, it may not be precise enough:{*current expression*}. Answer the following questions. {*Questions generated by Real-LOD*}.
>
> **Visual prompt**
> Visual prompts of three actions for VLM:
>
> | *object itself* | *object and surrounding areas* | *whole image* |
> |:---:|:---:|:---:|
> | **Object Crop** | **Extended Object Crop** | **Object Highlight** |

Table 12: Visual and language prompts for VLM tools. We show examples of the three image editing actions for VLM. Visual and language prompts are generated case by case via our Real-LOD.

## H ANALYSIS ON COMPUTATION COST

In this section, we present statistical results to analyze the computation cost of our Real-LOD, which are referred to by Sec 4.3. Tab. 13 is the average number of calls for each step and the time cost during each call for one expression, and Tab. 18 presents the distribution of iteration number in Real-LOD. Note that the max iteration number here is set to 10 for investigation.

Table 13: Average number of calls for each step and time cost during each call for one expression.

| Step | Avg num of calls | Time cost of each call |
|---|---|---|
| Planning | 3.09 | 0.265s |
| LLM-tool | 0.65 | 0.131s |
| VLM-tool | 0.43 | 0.159s |
| Reflection | 2.08 | 0.291s |

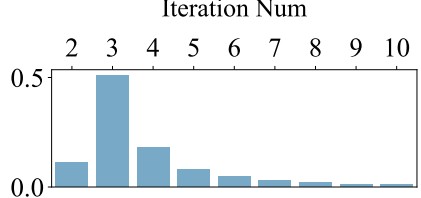

Figure 18: Distribution of iteration number in Real-LOD. Note that the max iteration number here is 10 for investigation.

# I    TECHNICAL DETAILS

## I.1    LICENSES OF DATASETS, CODES AND MODELS

In Tab. 14, we present the Licenses and URLs of datasets, codes and models used in our paper.

Table 14: The License and URL of datasets, codes and models utilized in this paper.

| Assert | Type | License |
|---|---|---|
| O365 (Shao et al., 2019) | Dataset | Creative Commons Attribution 4.0 License. |
| OpenImage (Kuznetsova et al., 2020) | Dataset | - |
| LVIS (Gupta et al., 2019) | Dataset | Creative Commons Attribution 4.0 License. |
| OmniLabel (Schulter et al., 2023) | Dataset | MIT License. |
| DOD (Xie et al., 2023) | Dataset | Creative Commons Attribution 4.0 License. |
| OVDEval (Yao et al., 2024) | Dataset | Apache-2.0 license. |
| Refcoco/g/+ (Mao et al., 2016) | Dataset | Apache-2.0 license. |
| MMDetection (Chen et al., 2019) | Code | Apache-2.0 license. |
| ChatGLM (Zeng et al., 2023) | Code | Apache-2.0 license. |
| LLaVA (Liu et al., 2023a) | Model | Apache-2.0 license. |
| Vicuna (Zheng et al., 2023) | Model | Llama 2 Community License Agreement. |

## I.2    TRAINING DETAILS OF REAL-MODEL

The implementation of Real-Model is based on the MMDetection (Chen et al., 2019) framework and PyTorch (Paszke et al., 2019). The input size of all the experiments is $1333 \times 800$, and the batch size is 4 per GPU. In the ablation study, there is only a single machine with 8 NVIDIA V100 GPUs for training to guarantee impartiality. For the final result, we train on 16 NVIDIA V100 GPUs for better performance. During training, we employ the AdamW optimizer (Kingma & Ba, 2015) with a momentum of 0.9 and a weight decay of 0.05. The learning rate setting includes a 1000-iteration warm-up with a start factor of $0.1$ and a multi-step schedule with an initial value of $4 \times 10^{-6}$ for 10 epochs. To be specific, the weights used for model initialization are referenced from the office repository of mm-GDINO (Zhao et al., 2024).

## I.3    EVALUATION DETAILS

In Tab. 15, we provide the detailed training data information of other LOD methods, which we compare within the OmniLabel and DOD benchmark.

Table 15: A detailed list of training data information for other LOD methods.

| LOD method | Backbone | Source | #Img |
|---|---|---|---|
| MDETR (Kamath et al., 2021) | ENB3 | COCO, RefC/g/+, VG, GQA, Flickr30k | 0.3M |
| GLIP (Li et al., 2022) | Swin-L | O365, COCO, OI, VG, ImageNet, GoldG, CC3M, CC12M, SBU | 17.5M |
| FIBER (Dou et al., 2022) | Swin-B | COCO, CC3M, SBU, VG | 4M |
| OWL-V2 (Minderer et al., 2023) | ViT-L | WebLI | 10B |
| UniNext (Yan et al., 2023) | ViT-H | O365, RefC/g/+ | 0.7M |
| UniNext (Yan et al., 2023) | ResNet-50 | O365, RefC/g/+ | 0.7M |
| GDINO (Liu et al., 2024b) | Swin-B | O365,OI,GoldG,CC4M,COCO, RefC/g/+ | 5.8M |
| OFA-DOD (Xie et al., 2023) | ResNet-101 | CC12M, CC3M, SBU, COCO, VG, RefC/g/+ | 16M |
| APE-A (Shen et al., 2024) | ViT-L | COCO, LVIS, O365, OI, VG | 2.0M |
| APE-B (Shen et al., 2024) | ViT-L | COCO, LVIS, O365, OI, VG, RefC/g/+ | 2.6M |
| mm-GDINO (Zhao et al., 2024) | Swin-T | O365, GoldG, GRIT, V3Det | 2.8M |
| mm-GDINO (Zhao et al., 2024) | Swin-B | GoldG, O365, COCO, OI, RefC/g/+, V3Det, LVIS, GRIT | 12M |

## I.4    MODEL DETAILS

In Fig. 19, we present more architectural details of the Real-Model, which is based on the mm-GDINO (Zhao et al., 2024). As shown in Fig. 19, the text encoder and image encoder first extract the text and image features, respectively. The bidirectional feature enhancement module is then used to integrate the text and image features through cross-modality cross-attention. After integration,

cross-modality queries are extracted from the image features with the language-guided query selection module and then subsequently input into the decoder with a further cross-modality fusion. The final output queries are then utilized for contractive loss and localization loss. More details can be found in (Yang et al., 2023b; Zhao et al., 2024).

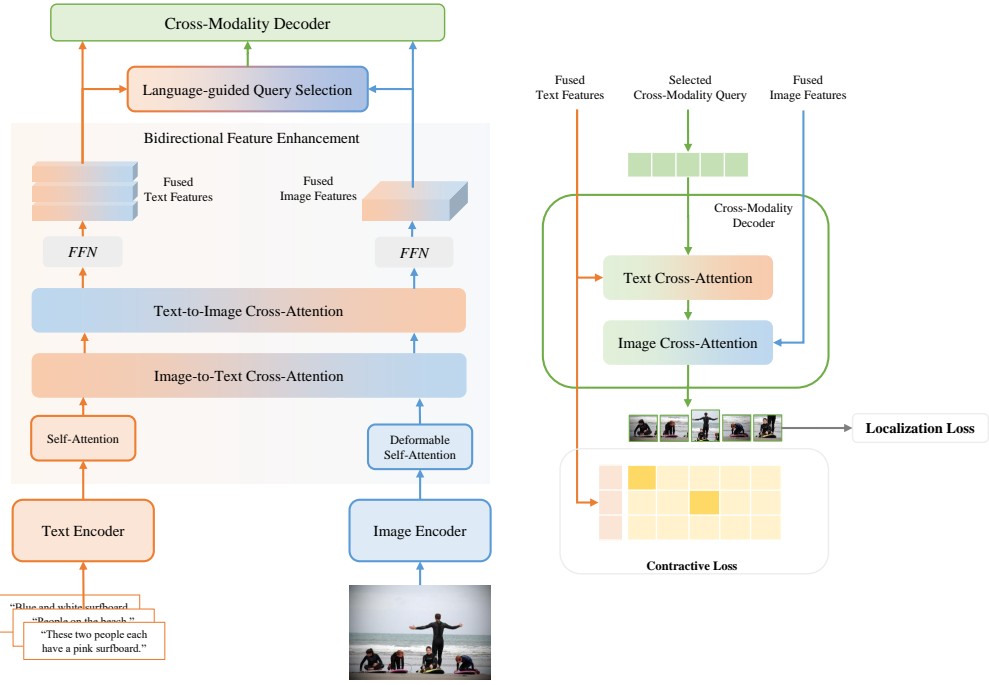

Figure 19: More architectural details of the Real-Model. The Real-Model is built upon the mm-GDINO (Zhao et al., 2024) and trained with Real-Data re-aligned with Real-LOD. The right part is the overall framework before the cross-modality decoder. The left part is the cross-modality decoder and loss calculation.

## J  DISCUSSION

### J.1  LESSONS OF AGENTIC WORKFLOW DESIGNING

**States/actions of agentic workflow.** The design of agentic workflow states/actions should follow the neural-symbolic spirit, guided by task-specific requirements analysis. In Real-LOD, we establish 5 core states/actions through systematic analysis of hallucinations brought by VLMs to produce expressions. These carefully designed actions, including re-perceive ROI areas and customized questions, are controlled by our agent for VLM to adaptively deal with different types of hallucinations.

**Agent selection suggestion.** During the implementation of our Real-LOD, we empirically found that LLM reasons are more accurate in pure language form, especially when facing long text prompts and cases requiring strong logic. In contrast, the language-generative VLM can not effectively reason when perceiving language and visual data simultaneously, despite their inherent strengths in visual content comprehension. A showcase of this ineffectiveness is the model hallucination that produces incorrect language expressions, as discussed in the second paragraph of Sec. 3.3. This ineffectiveness is due to the naturally inadequate alignment between image and language data used for VLM training (*e.g.,* it is difficult to fully represent what an image conveys in 1-2 sentences). Therefore, we use VLM to convert visual content to descriptive text and choose LLM with pure language form as the agent to leverage its strong logical reasoning ability for accurate and robust planning within our workflow.

**Training data to fine-tune agent.** Regarding the training data to fine-tune the agent, it does not necessarily include all the recurrent steps. This is because the planning process of the agent in each

recurrent step is the same, and we only need to focus on the behavior of the agent in one recurrent step.

## J.2 BROADER IMPACT

Agent and language-based object detection have shown significant applications in various real-world scenarios, particularly in intelligent robotics and autonomous driving. Our proposed method exhibits potential for these two research areas, offering valuable insights to the community. Our method focuses on correcting language expressions for the LOD dataset without a specific application goal. Hence, it does not directly involve societal issues.

## J.3 LIMITATION

Real-LOD employs the VLM to perceive the content of a given target in various scenes, providing external information to help the linguistic descriptions correcting process for reducing the model hallucination. Although our method strongly stimulates the potential of VLMs by introducing agentic workflows and visual tools, there are still some unmanageable hard cases limited by the original performance of VLMs. As shown in Fig 20, there are two main kinds of data refinement error caused by the error perception of VLM:

1. Typical visual hard cases. Object-detection datasets include low-light or low-quality scenes and extremely small or difficult-to-recognize objects. It could be difficult for VLM to generate appropriate expressions for these targets.

2. Expression describes a foreground object instead of the target background object. VLM may ignore the target object in the background when an occlusion exists. In order to ensure high quality at the bbx level, the reflection module regards these expressions as wrong.

When conducting large-scale data refinement, we set the maximum iteration to 4. With several extra iterations, some failure cases are likely to be solved. Since we already have a large amount of high-quality data for downstream training with a task-solved rate of 75%, we choose not to increase this parameter for the sake of efficiency. This indicates the requirement of developing more powerful and robust VLMs to handle complicated situations more efficiently.

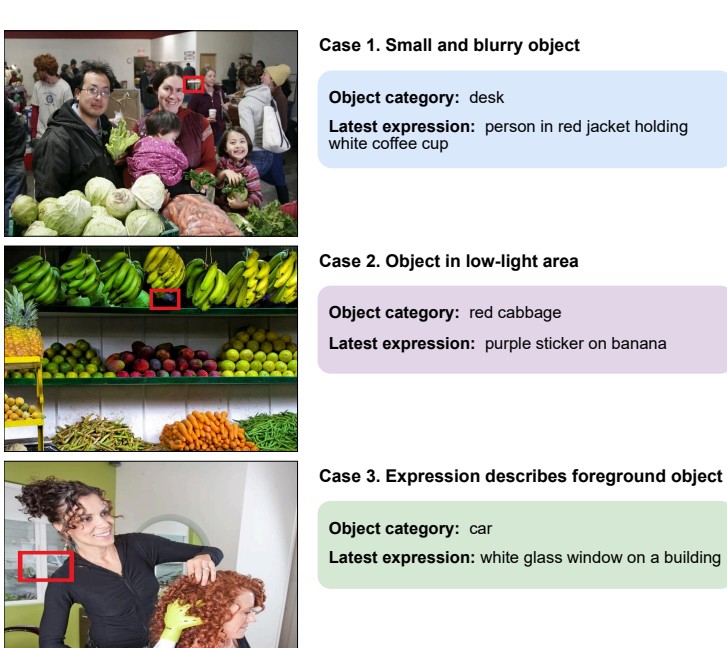

Figure 20: Visualization of some failure cases.

## K    MORE ANALYTICAL EXPERIMENT

In this section, we provide additional analytical experiments about our Real-LOD to further demonstrate its effectiveness. Tab. 16 reports the accuracy in choosing the corresponding state/action on the validation set of fine-tuning data (1k samples). It indicates that Real-Agent can accurately reason the state/action, especially for the "Wrong" and "Uncertain" states. In addition, the relatively lower accuracy of the "Correct" state indicates that our agent is strict with the quality of expression to prevent the hallucinations as much as possible. According to Tab. 17, we provide a more detailed ablation study of Real-LOD. The experimental setup can be found in Sec. 3.3. The "w/o Planning" is the same as the random selection schema in Sec. 3.3. "w/o Cyclic Workflow" indicates the workflow with only one cycle. The results intuitively illustrate the importance of each component to our agentic workflow.

Table 16: Accuracy in choosing the corresponding state/action on the validation set of fine-tuning data.

| State/Action | Accuarcy |
|---|---|
| Correct/Stop | 93.1% |
| Wrong/LLM | 99.4% |
| Uncertain/Object Crop | 99.6% |
| Uncertain/Extended Object Crop | 95.8% |
| Uncertain/Highlight | 90.0% |

Table 17: More detailed ablation study of our agentic workflow.

| State/Action | Success Rate |
|---|---|
| Real-LOD | 74.7% |
| w/o Planning | 35.6% |
| w/o Action 2 | 18.0% |
| w/o Action 3 | 53.0% |
| w/o Action 4 | 51.8% |
| w/o Action 5 | 57.4% |
| w/o Cyclic Workflow | 60.7% |

