# OpenReview forum: "Re-Aligning Language to Visual Objects with an Agentic Workflow"
_ICLR.cc/2025/Conference — ICLR 2025 Poster_

### Official Review · Reviewer_M1hG · 2024-10-30

**Soundness:** 3
**Presentation:** 2
**Contribution:** 3
**Rating:** 6
**Confidence:** 5

**Summary:**

This paper begins by explaining that directly using vision-language models (VLMs) for re-captioning target regions can lead to hallucinations, highlighting that VLMs alone may not be suitable for training data optimization. To address this, the authors propose an Agent Workflow (Real-LOD) to improve training sets for open-set object detection, which consists of Planning, Tool Usage, and Reflection stages. This workflow combines the reasoning ability of LLM (ChatGLM-6B) and the perception ability of VLM (LLaVAv1.6-34B) to effectively correct inaccurate object descriptions within the training dataset.  The authors then re-train an open-set object detector based on the Swin-B backbone using the re-aligned dataset. This detector performs well across multiple open-set detection benchmarks, including the new challenging OmniLabel benchmark. Extensive ablation studies on these benchmarks further validate the effectiveness of each step in the data refinement process.

**Strengths:**

1. **Innovative Data Optimization Method**: The proposed Agent Workflow offers an innovative approach to training data refinement by employing multi-step processes—namely, Planning, Tool Usage, and Reflection—to mitigate hallucinations generated by Vision-Language Models (VLMs) and enhance data quality.

2. **Demonstrated Effectiveness in Experiments**: The retrained open-set object detector achieves strong performance across challenging benchmarks, including OminLabel, underscoring the practical efficacy of this approach.

3. **Relatively Comprehensive Ablation Studies**: The paper includes extensive ablation studies that validate the necessity and contribution of each step within the Agent Workflow, providing robust empirical support and reinforcing the reliability of the study.

4. **Clear and Informative Visualizations**: The figures are well-designed, effectively illustrating the issues with VLMs and detailing the steps within the Agent Workflow, enhancing reader understanding of the methodology.

**Weaknesses:**

1. **Choice of VLM**: The authors use the commonly utilized large-parameter version of VLM (LLaVAv1.6-34B). However, several newer Controllable VLMs / MLLM / Captioner allow interaction through a variety of visual prompts (such as Points, Boxes, and Masks) to obtain more focused descriptions for target regions. Directly using these controllable VLMs may have fewer hallucinations on region caption task and may be better than LLaVAv1.6-34B with region cropping. Examples Region Controllable VLMs include: AlphaCLIP (CVPR 2024) [1], Osprey (CVPR 2024) [2], Ferret (ICLR 2024) [3], OMG-LLaVA (NeurIPS 2024) [4] and so on.
2. **Lacking ablation experiments on the influence of the fine-tuning of ChatGLM**. What is the role of this step? Can fine-tuning enhance the effect of all three stages of the Agent.
3. **Error analysis about the workflow of agent**. Lack of analysis of Agent workflow failure cases. To which part of the Agent should the cause of the data refinement error be attributed: inference? Tool call? Error perception of the tool itself? Error of reflection?
4. **Computation cost is two large**, 48 V100 GPUs, Normal LABS can't afford it. A lite version should be studied in the future.
5. **Minors**:
    1. Some expressions are more obscure, such as **Re-Aligning Language to Visual Objects**, so it is better to directly use the refine object description in open-v object detection data, which is more clearly and easy to understand.
    2. The use of symbols and naming are not easy to distinguish. For example, the base name $\text{Real LOD}$ used three times, $\text{Real LOD}^{data}, \text{Real LOD}^{model}, \text{Real LOD}^{agent}$. This discourages the reader from quickly distinguishing between these symbols and leads to confusion.

[1] Sun Z, Fang Y, Wu T, et al. Alpha-clip: A clip model focusing on wherever you want[C]//Proceedings of the IEEE/CVF Conference on Computer Vision and Pattern Recognition. 2024: 13019-13029.

[2] Yuan Y, Li W, Liu J, et al. Osprey: Pixel understanding with visual instruction tuning[C]//Proceedings of the IEEE/CVF Conference on Computer Vision and Pattern Recognition. 2024: 28202-28211.

[3] You H, Zhang H, Gan Z, et al. Ferret: Refer and ground anything anywhere at any granularity[J]. arXiv preprint arXiv:2310.07704, 2023.

[4] Zhang T, Li X, Fei H, et al. Omg-llava: Bridging image-level, object-level, pixel-level reasoning and understanding[J]. arXiv preprint arXiv:2406.19389, 2024.

**Questions:**

Above all this is a good idea. Refining training data to gain more high-quality data for training the model is highly practical. However, I have a question about whether the region-controllable captioner is better than a complex agent workflow. The former is more cheap and easy to use.

---

> ### Author Response · Authors · 2024-11-23
> **Response to Reviewer M1hG (1/2)**
>
> Dear Reviewer M1hG,
>
> We would like to thank you for providing valuable comments, and we answer the concerns raised below. We have included all discussion results in our revised manuscript (content in orange color).
>
> > Choice of VLM: The authors use the commonly utilized large-parameter version of VLM (LLaVAv1.6-34B). However, several newer Controllable VLMs / MLLM / Captioner allow interaction through a variety of visual prompts (such as Points, Boxes, and Masks) to obtain more focused descriptions for target regions. Directly using these controllable VLMs may have fewer hallucinations on region caption task and may be better than LLaVAv1.6-34B with region cropping. Examples Region Controllable VLMs include: AlphaCLIP (CVPR 2024) [1], Osprey (CVPR 2024) [2], Ferret (ICLR 2024) [3], OMG-LLaVA (NeurIPS 2024) [4] and so on.
>
> We appreciate the valuable advice. We observe that controllable VLMs may have hallucinations on region caption tasks like LLaVAv1.6-34B with region cropping. Our proposed method is a generalized workflow schema compatible with these different VLM tool models and various visual prompts (such as Points, Boxes, and Masks). The goal of these visual prompts is the same as our visual prompts updated by image editing operations (especially the object highlight operation) to help VLM focus on different ranges of target regions. In addition, the adaptive customization of text and visual prompts generated by the agent is one of the core contributions of our workflow that can also help different VLMs perform better. We chose the LLaVAv1.6-34B and those simple image editing operations because they are widely used and can better demonstrate the generality of our method. These strong VLMs with better detail reasoning capabilities encourage us to explore their ability further to help our workflow deal with more complex cases in the future. We have cited them in Ln 45-47 to show our respect.
>
> &nbsp;
>
> > Lacking ablation experiments on the influence of the fine-tuning of ChatGLM. What is the role of this step? Can fine-tuning enhance the effect of all three stages of the Agent.
>
> The role of fine-tuning is to adapt ChatGLM into a specific agent that can plan states/actions and output results in a desired format in our workflow for VL (vision-language) data alignment while preserving the reasoning ability. The fine-tuning strongly enhances the effect of all stages of the agent. When directly using the pure ChatGLM, the planning process is uncontrollable and can not reason for the desired output in a specific format, which our workflow can not parse. This drops the success rate to a low degree, nearly the success rate of random selection schema, as shown in Sec. 3.3.
>
> &nbsp;
>
> > Error analysis about the workflow of agent. Lack of analysis of Agent workflow failure cases. To which part of the Agent should the cause of the data refinement error be attributed: inference? Tool call? Error perception of the tool itself? Error of reflection?
>
> Thanks for pointing out the lack of analysis of failure cases. It has a high guiding significance for further optimization of Real-LOD. We look into the failure cases and find out there are two main kinds of data refinement error caused by the error perception of the tool model (VLM):
>
> 1. Typical visual hard cases. As illustrated in Sec I.2 of the appendix, object-detection datasets include low-light or low-quality scenes and extremely small or difficult-to-recognize objects. It could be difficult for VLM to generate appropriate expressions for these targets.
> 2. Expression describes a foreground object instead of the target background object. VLM may ignore the target object in the background when an occlusion exists. In order to ensure high quality at the bbox level, the reflection module regards these expressions as wrong.
>
> When conducting large-scale data refinement, we set the maximum iteration to 4. With several extra iterations, some failure cases are likely to be solved. Since we already have a large amount of high-quality data for downstream training with a task-solved rate of 75%, we choose not to increase this parameter for the sake of efficiency.
>
> Revised contents:
> 1. We have included visualizations and analysis of failure cases in Sec. I.2 of the appendix.

---

> > ### Author Response · Authors · 2024-11-23
> > **Response to Reviewer M1hG (2/2)**
> >
> > > Minors:
> > > - Some expressions are more obscure, such as Re-Aligning Language to Visual Objects, so it is better to directly use the refine object description in open-v object detection data, which is more clearly and easy to understand.
> > > - The use of symbols and naming are not easy to distinguish. For example, the base name used three times, . This discourages the reader from quickly distinguishing between these symbols and leads to confusion.
> >
> > We appreciate these two valuable suggestions. For the first suggestion, our paper focuses on the LOD task[1], which is more comprehensive than the OVD task, and the motivation of our paper is not only to refine pure-text description but also to achieve a better alignment of language description to corresponding visual objects, which is also the core proposal of the LOD task. For the second suggestion, we have replaced the Real-LOD$^{agent}$, Real-LOD$^{model}$, and Real-LOD$^{data}$ with Real-Agent, Real-Model, and Real-Data.
> >
> > [1] Schulter et al., OmniLabel: A Challenging Benchmark for Language-Based Object Detection. ICCV, 2023.

---

### Official Review · Reviewer_e1U7 · 2024-10-31

**Soundness:** 2
**Presentation:** 3
**Contribution:** 1
**Rating:** 5
**Confidence:** 3

**Summary:**

The paper aims at the language-based object detection task that detect visual objects based on language queries. Since previous vision-language models (VLMs) may face the hallucination challenge, authors present an agentic workflow based on an LLM to re-align the language query and visual objects. The workflow contains three main steps: planning, tool use, and reflection. Some evaluation results on some benchmarks show the performance comparison.

**Strengths:**

Figure and tables are clear. Many figures and tables are used to present the language expressions and evaluation results.

A state-of-the-art method mm-GDINO (Zhao et al., 2024) is used for performance comparison.

In Figure 5, the detail of how the presented Real-LOD method re-aligns one raw language query to given image is clear. Figure 5 can help reader understand the model implementation.

Ablation study on the Omnilabel dataset is conducted for model anaysis.

**Weaknesses:**

Authors use an agentic workflow controlled by an LLM to reduce VLM hallucinations. However, there is still hallucinations in the LLM. In fact, hallucinations are not effectively reduced.

The general LOD framework in Figure 3 is very common. Thus, I have concern about the technological novelty of the presented method.

In Section 3.2, there are "Right" state and "Wrong". How to judge whether he is truly right or wrong, rather than the hallucinations from the LLM?

The works seems to directly use the LLM and VLM. There is no real innovation for both LLM and VLM.

**Questions:**

I hope that authors can address our questions and concerns in Weaknesses.

---

> ### Author Response · Authors · 2024-11-23
> **Response to Reviewer e1U7**
>
> Dear Reviewer e1U7,
>
> We would like to thank you for providing valuable comments, and we answer the concerns raised below. We have included all discussion results in our revised manuscript (content in orange color).
>
> > - Authors use an agentic workflow controlled by an LLM to reduce VLM hallucinations. However, there is still hallucinations in the LLM. In fact, hallucinations are not effectively reduced.
> > - In Section 3.2, there are "Right" state and "Wrong". How to judge whether he is truly right or wrong, rather than the hallucinations from the LLM?
>
> We agree that there are still hallucinations in the LLM. However, we have conducted the following schemes to help the agent judge whether the expression is truly right or wrong (i.e, VLM hallucination), rather than the hallucinations from the LLM:
>
> 1. We introduce the ground truth category label from the detection dataset. This label serves as a strong reference during reflection. Our Real-LOD can identify the expression that contradicts the label as a VLM hallucination (i.e., truly wrong).
> 2. We observe that different foundation models (i.e., Vicuna, Llava) in our workflow normally do not encounter the same hallucinations. Therefore, we use them to interact, examine, and improve expressions iteratively within the workflow. This enables the Real-LOD to judge whether the current expression is truly wrong from diverse perspectives, advancing its robustness and reducing hallucinations from a single model. Meanwhile, when model discrepancy still exists once reaching the max iteration number, we regard this as a hallucination and can directly discard this expression to preserve training data quality. Besides, our specifically designed prompts for these models also further reduce their own potential hallucinations.
> 3. We introduce an LLM-based reflector in each iteration to analyze the correctness of the current expression based on the given reference information. We observed that the reflector never introduces new errors caused by LLM hallucinations and has a high recall for expressions that may have a VLM hallucination (i.e., Uncertain/Wrong states). This is because text analysis is the primary strength of LLMs, and the reflection task is straightforward. Therefore, its feedback to the agent can help Real-LOD further reduce the VLM hallucinations in a given expression.
>
> With the above schemes, the hallucinations are effectively reduced, demonstrated by the improvement of the evaluation results in Sec. 4 and examples in Fig. 5 and Sec. D.
>
> &nbsp;
>
> > The general LOD framework in Figure 3 is very common. Thus, I have concern about the technological novelty of the presented method.
>
> The general LOD framework in Fig. 3 is not our contribution but to introduce the background task and how paired VL (vision-language) inputs predict target objects. The technological novelty of our paper is that we pioneer the agentic workflow by designing reasoning/planning, tool use, and reflection steps as a whole pipeline to improve VL alignment from a data-centric perspective. In addition, the states/actions are specifically designed based on our data analysis of visual objects and corresponding language expressions. The improved VL data continuously benefit all LOD methods even though their model architectures evolve.
>
> &nbsp;
>
> > The works seems to directly use the LLM and VLM. There is no real innovation for both LLM and VLM.
>
> While we use the LLM and VLM as our tool models in the workflow, we would like to emphasize that our work is not a direct application of them, and the innovation resides not in the model itself but in the design of the proposed agentic workflow. Our design rationale formulates different models (LLM and VLM) as a whole system through the coherent workflow rather than an individual off-the-shelf integration. The following are three aspects of our innovation:
>
> 1. We pioneeringly develop a coherent agentic workflow containing planning, tool use, and reflection steps for VL data alignment. We carefully design each step to ensure the overall performance.
> 2. Our state/action design is based on our VL data analysis in Ln 355-364, where we first categorize hallucination types and develop actions accordingly. Fig. 6 and 7 show that our actions effectively refine raw expressions.
> 3. Our workflow pioneeringly improves data quality from the VL alignment perspective. Our data-centric design benefits all the LOD model training regardless of their model architecture evolvement.
>
> Besides, we empirically find that adding global image captions (e.g., Fig. 5) further improves the reasoning ability in our planning step. This is because we find out the LLM reasons more effectively in pure language rather than VL form. Moreover, we utilize LoRA tuning (i.e., Ln 344) to maintain the original reasoning ability of LLM. As such, these efforts indicate that we intend to integrate different models more coherently through agent characteristics rather than directly using them.

---

> > ### Comment · Reviewer_e1U7 · 2024-12-03
> >
> > After reading the responses and other reviews, I only concern the writting. Thus, I want to keep my rating.

---

> > > ### Author Response · Authors · 2024-12-04
> > > **Thanks for your reply!**
> > >
> > > Dear Reviewer e1U7,
> > >
> > > Thanks for the continued engagement! We are glad our responses have addressed your initial concerns about novelty and hallucinations. As for the writing concern, following the suggestions of Reviewers (i.e., LXzJ, J3CB, and M1hG), we have modified our paper point by point to clarify the unclear contents, further improving the readability of our paper. The revised contents are in orange color. We hope this can address your concern.
> > >
> > > &nbsp;
> > >
> > > Best regards,
> > >
> > > The Authors

---

> ### Author Response · Authors · 2024-11-28
> **Sincerely Look Forward to Your Feedback!**
>
> Dear Reviewer e1U7,
>
> We hope this message finds you well.
>
> Thanks again for your insightful suggestions and comments. As the author-reviewer discussion time is limited, we want to ensure that we address any remaining uncertainties or questions you may have.
>
> After carefully considering your comments, we have taken the necessary steps to provide additional clarifications, particularly how our workflow reduces hallucinations and the novelty of our method. We hope that our explanations and additional details contribute to a clearer understanding of our workflow pipeline.
>
> We are open to providing further clarifications or conducting additional experiments if needed. Please feel free to reach out to us with any further inquiries.
>
> Once again, we are thankful for your time and attention to our work.
>
> &nbsp;
>
> Best regards,
>
> The Authors

---

> ### Author Response · Authors · 2024-12-03
> **Sincerely Look Forward to Your Feedback Again!**
>
> Dear Reviewer e1U7,
>
> As the author-reviewer discussion time is one day left, we sincerely request a reply from you again.
>
> We have carefully responded to your comments: 1. Regarding the question about novelty, which Reveiwer LXzJ, J3CB and M1hG have acknowledged, we have provided a detailed explanation to reemphasize the novel contribution of our method. We have also clarified that Fig. 3 is not our contribution but to introduce the background task. 2. Regarding the hallucination problem, we have further illustrated the three detailed schemes to solve it in our workflow.
>
> We would like to ensure that the above responses have answered your uncertainties or questions. We are open to providing further clarifications or conducting additional experiments if needed. We sincerely look forward to your reply with any additional inquiries.
>
> Once again, we genuinely appreciate your time and effort in our work.
>
> &nbsp;
>
> Best regards,
>
> The Authors

---

### Official Review · Reviewer_LXzJ · 2024-11-01

**Soundness:** 2
**Presentation:** 1
**Contribution:** 3
**Rating:** 5
**Confidence:** 4

**Summary:**

The authors of this paper propose a method for improving the quality of text descriptions used for training language-based object detection (LOD) models. As done in prior work, in the first step, they use a vision-language model with language generation capabilities (LLava in their case) to generate text descriptions for objects.

Their main contribution is a multi-step iterative method for improving the generated descriptions for each object. The authors do a manual analysis and define six different problems that can occur for each description.  They group these problems into three categories (referred to as states) and pre-define an action for each problem (e.g., if the attributes in the description are not correct, crop the image and generate a new description).

In each iteration of their method, a fine-tuned unimodal LLM (i.e., their agent) inspects the description and decides if the description is good enough or needs further modification. If further modification is needed, the agent selects one of the three predefined states (i.e., types of problems) and suggests taking the corresponding action to refine the description or generate a new one.
Once the action is complete, the authors use another LLM to reflect on the output of the executed action. Although it is not entirely clear in the paper how it is accomplished, the results of the reflection process are integrated into the next iteration to further improve the newly generated or updated object description. The authors repeat the iterations for each description for a certain number of steps or terminate the loop if the agent decides the description is good enough and no further modification is needed.

**Strengths:**

* The idea that several models work together to collectively improve the quality of the training data is very interesting and could be potentially useful for other applications as well.
* The authors follow a reasonable design process to develop their method. For example, they define the set of states and corresponding actions based on the analysis of the existing data.
* The authors ask the right questions when evaluating their method. For example, they attempt to study the importance of correctly choosing the state/actions for each description. They also try to isolate the impact of higher-quality data from the original data that they are using (paragraph 2 of Section 4.1).

**Weaknesses:**

I have two major concerns about this work. Although the idea is interesting, and the authors have come up with the correct questions, the quality of the writing and experimental setup can be improved significanlty.

**Writing:**

The paper is not well written and is hard to understand. It takes a couple of passes to understand the arguments. There are also a lot of ambiguities that need further clarification. Here are some examples:
* Second paragraph of section 3.1: Since the authors are generating the initial training data for LOD (that they will later re-align with Real-LOD) and not using the data that is out there from previous work, the setup and how the data generation is accomplished should be clearly explained. The authors just mentioned the name of the model they use.
  * For the initial expression generation with LLava, the authors should clearly state the prompts used, formatting structure, number of expressions generated, etc.
  * The authors also mention, “vicuna is used to diversify the generated descriptions”. What does “diversify” mean in this context? How exactly is it done, given the generated data?
* Since the authors are fine-tuning “ChatGLM-6B” as their agent (which is responsible for choosing the state and action to be executed), it seems like the agent does not have access to the image, and it reasons only based on the textual prompts. However, the figures and texts give the impression that the image is also used as input to the agent to choose the state/action. This should be clearly stated in the text that the agent does not have access to the image.
  * Also related, it seems like the agent does not use the bounding box location either. This should also be clarified.
* Line 260: Vague statement: “When using these models, Real-LOD agent adaptively modulates visual content and text prompts based on its reasoning thoughts”. It is not clear which aspect of the agent this statement is referring to.
* How is the output of the reflector incorporated in the next iteration of Real-LOD? If, in the next iteration, the agent decides that no further modification is needed, is the reflector output not used?
* The authors should discuss the expected behavior of the planner in more detail in the main text. For example, does it format the questions (i.e., prompts) that are used with other tools (i.e., LLM and VLM) in free form, or is it expected to follow specific guidelines/templates for generating the prompts?
* Section 3.3 needs more organization. The authors should clearly separate the analysis process, the statistics of the created data for training real-LOD-agent, and the analytical experiment to test the quality of the generated data.
* Line 340: SigLip filtering here seems to be different from the one explained in Line 197. It is not clear what is being filtered here and how the filtering is accomplished.
* Line 341: “Each input data contains the object category label, raw expression, and reasoning from LLM.” what reasoning from LLM? How is it generated? My impression is that this is the input to train the agent. So, we do not have the agent yet to get the reasoning as explained in other parts of the paper. So, this step should be further clarified.
* Line 341-342: “Then, we manually set the state/action and collect responses as the outputs.” Is the manual annotation done for all 15k input samples? If yes, it should be reported who has done the labeling (i.e., authors themselves or mechanical Turk, etc.), and the details of the instructions given to labelers should be provided.
* Line 350: “We randomly select hundreds of filtered expressions and manually check each for a detailed observation”. This is an important detail of the analysis. “Hundreds” should be quantified with exact numbers.
* Line 379: “we find that the success rate is 74.7% v.s. 35.6%.” these numbers and the experimental setup used to achieve these are not clear.
* it is mentioned in the abstract that the method results in "50% improvement". Does it mean that the proposed method improved the performance by 50 percentage points? that does not seem to be the case in the experiments.
* It is also more informative if the authors report the average improvement rather the maximum.

**Minor suggestions:**

* It might be easier to understand if section 3.3 (DATA ANALYSIS OF LANGUAGE AND VISUAL OBJECTS) comes before section 3.2 (AGENTIC WORKFLOWS FOR LANGUAGE EXPRESSION RE-ALIGNMENT)
* Similarly, it might be helpful to have section 4.2 ( COMPARISONS WITH STATE-OF-THE-ART LOD METHODS) before section 4.1 (ABLATION STUDY)

**Experiments**

* The authors claim significant boosts in performance but the experiments are very limited (only three datasets). More extensive experiments are needed to show the merits of the proposed method.
* For section 3.3, although authors have done the analysis that shows choosing states and actions by their agent achieves better performance than random selection of states/actions. It is important to also report the accuracy of the agent in choosing the corresponding state/action for each input sample.
* Correct me if I am wrong, but it seems like the agent is only trained on data built from Objects365. It would be interesting to evaluate the model on Objects365 dataset exclusively to analyze how transferable the agent is to other datasets. Although object365 is included in Omnilabel but exclusive numbers for Objects365 help better understand the impact of distribution shift from training to inference.

I also have some questions about the design choices:
* Why use an LLM as the agent when the input contains both image and text? Isn’t a language-generative VLM a better choice?
* Similarly, in state2/action2, why use an LLM to write a new object description based on the existing caption rather than using a VLM to write the description based on both the image and the caption?

**Questions:**

Please see the weaknesses.

---

> ### Author Response · Authors · 2024-11-23
> **Response to Reviewer LXzJ (1/5)**
>
> Dear Reviewer LXzJ,
>
> We would like to thank you for providing valuable comments, and we answer the concerns raised below. We have included all discussion results in our revised manuscript (content in orange color).
>
> > Second paragraph of section 3.1: Since the authors are generating the initial training data for LOD (that they will later re-align with Real-LOD) and not using the data that is out there from previous work, the setup and how the data generation is accomplished should be clearly explained. The authors just mentioned the name of the model they use.
> >- For the initial expression generation with LLava, the authors should clearly state the prompts used, formatting structure, number of expressions generated, etc.
> >- The authors also mention, “vicuna is used to diversify the generated descriptions”. What does “diversify” mean in this context? How exactly is it done, given the generated data?
>
> We appreciate the valuable suggestions and make the clarification here. Since the initial expression generation before re-aligning is not the main contribution of our paper, we simplify the presentation and provide an overview of the whole pipeline in the Sec. A of the appendix where are two steps to generate initial language expressions.
>
> 1. Regarding the details of expression generation through VLM, we use LLaVA to generate 673k expressions for 188k images with 336.5k objects, following previous works [1, 2]. For each object, we randomly select two prepared prompts (shown in Table 8 of Sec. G.1) with an image and corresponding category for LLaVA to generate expressions.
> 2. As for the second point, "diversify" means generating synonyms to expand the richness of language expressions. Vicuna is introduced to achieve this proposal by redescribing the given expression, and it expands the number from 673k to 1,346.1k. The prompt is shown in Table 8 of Sec. G.1. We repeat the process two times for each expression.
>
> Revised Contents:
> 1. In the second paragraph of Sec. 3.1, we have modified the uncertain part and added the reference to Sec. A.
> 2. In the Sec. A, we have added a detailed explanation of the initial expression generation.
> 3. In the Table 8 of Sec. G.1, we have provided the details of the prompt template.
> 4. We replace the "initial" with "raw" to keep the consistency of the paper.
>
> [1] Dang et al., Instructdet: Diversifying referring object detection with generalized instructions. ICLR, 2024.
>
> [2] Pi et al., Ins-detclip: Aligning detection model to follow human-language instruction. ICLR, 2024.
>
> &nbsp;
>
> > Since the authors are fine-tuning “ChatGLM-6B” as their agent (which is responsible for choosing the state and action to be executed), it seems like the agent does not have access to the image, and it reasons only based on the textual prompts. However, the figures and texts give the impression that the image is also used as input to the agent to choose the state/action. This should be clearly stated in the text that the agent does not have access to the image.
> > - Also related, it seems like the agent does not use the bounding box location either. This should also be clarified.
>
> We appreciate the valuable suggestions and make some clarification on the agent input here:
>
> 1. We agree that the proposed agent does not have access to the image, and we have clarified this point in Ln 274-277 to address the confusion. The reason why we do not provide the agent with the image is that we find LLM reasonings are more accurate in pure language form than in VL (vision-language) form.
> 2. The bbox locations are indeed not used for the reasoning process of agent, which we have clarified in Ln 276. However, these locations are provided for specific tools in the tool use step. They help LLM avoid positional errors in rewriting and enable VLM to focus on the object region during re-perception.
>
> Revised Contents:
> 1. We have clarified the above presentation to address the confusion of agent input in Sec. 3.2.
> 2. We have also provided more detailed information about the agent input in Sec. 3.2.
>
> &nbsp;
>
> > Line 260: Vague statement: “When using these models, Real-LOD agent adaptively modulates visual content and text prompts based on its reasoning thoughts”. It is not clear which aspect of the agent this statement is referring to.
>
> We apologize for the unclear statement. This statement refers to the capability of the agent to adaptively decide the visual content and text prompt for the chosen tool, i.e., the "Image editing" and "Prompt" parameters, as shown in Fig. 5. We have modified this statement in Ln 302 (i.e., Ln 260 in origin manuscript): "Based on its reasoning about the state of the current expression, Real-LOD$^{agent}$ adaptively modulates visual content and text prompts by setting up "Prompt" and "Image editing" parameters for scheduled tools. Then, the tool can be used effectively to get desired responses from VLM to improve the expression refinement.".

---

> > ### Author Response · Authors · 2024-11-23
> > **Response to Reviewer LXzJ (2/5)**
> >
> > > How is the output of the reflector incorporated in the next iteration of Real-LOD? If, in the next iteration, the agent decides that no further modification is needed, is the reflector output not used?
> >
> > Sorry for causing the misunderstanding. As presented in Ln 315, in the cyclic workflow, the analysis of reflector will be formulated as feedback to the agent, facilitating its planning in the next iteration. The agent takes the feedback as a strong reference which indicates the current refining quality to reason the state and arrange actions. Hence, in the next iteration, even though the agent decides that no further modification is needed, the reflector output is still needed to formulate this decision.
> >
> > &nbsp;
> >
> > > The authors should discuss the expected behavior of the planner in more detail in the main text. For example, does it format the questions (i.e., prompts) that are used with other tools (i.e., LLM and VLM) in free form, or is it expected to follow specific guidelines/templates for generating the prompts?
> >
> > We appreciate the suggestion and would like to clarify some points here. The agent (i.e., planner) follows specific guidelines to generate different prompts for different tool models (i.e., LLM and VLM), depending on their role and the state of given expressions. We illustrate two scenarios below:
> >
> > 1. Regarding the LLM for rewriting, the agent formulates the given information (i.e., image caption, expression, object category and location) to generate the in-context text prompt following the hand-craft template in Table 9.
> > 2. Regarding the VLM for visual content re-perception, the agent generates text and visual prompts following its reasoning about the state realized by the particular fine-tuning process.
> >     - Specifically, the text template (shown in Table 12) is formulated with a generated open-ended question related to uncertain information about the target object. For example, in Fig. 5, when the agent reasons that `'sand'` is uncertain, the question `'Is the horse walking on sand or on a dirt path?'` is generated to guide the VLM to get desired supplementary information.
> >     - The visual prompt is generated through one of the predefined image editing operations (shown in Table 12) to update the focus region for VLM. For example, in Fig. 5, when the agent reasons the uncertain `sand` is the surroundings of the `'horse'`, it selects the extended object crop to guide the VLM focus on the extended region that contains the target object and its surroundings.
> >
> > Revised content:
> > 1. We have provided details about the prompt of LLM-for-Rewrting in Ln 282 and updated Table 9 to be more apparent.
> > 2. We have provided details about the prompt of VLM in Ln 309.
> >
> > &nbsp;
> >
> > > Section 3.3 needs more organization. The authors should clearly separate the analysis process, the statistics of the created data for training real-LOD-agent, and the analytical experiment to test the quality of the generated data.
> >
> > We appreciate the valuable advice. We have updated Sec. 3.3 by organizing three subsections starting with "Training data for Real-LOD$^{agent}$", "Analysis of language and visual objects", and "Analytical experiment for Real-LOD".
> >
> > &nbsp;
> >
> > > Line 340: SigLip filtering here seems to be different from the one explained in Line 197. It is not clear what is being filtered here and how the filtering is accomplished.
> >
> > We appreciate the suggestion and clarify the usage of SigLip here. The SigLip filtering in Ln 323 (i.e., Ln 340 in origin manuscript) is the same as in Ln 213 (i.e., Ln 197 in origin manuscript). The filtering is accomplished by calculating the matching score between the visual object and the corresponding language expression through SigLip. Then, VL (vision-language) data pairs with matching scores lower than 0.5 are filtered out.
> >
> > Revised contents:
> > 1. In Sec. 3.3, we have provided the threshold of VL alignment score to filter expression in Ln 213.
> > 2. In Sec. 3.3, we have provided more details of SigLiP in Ln 323.

---

> > > ### Author Response · Authors · 2024-11-23
> > > **Response to Reviewer LXzJ (3/5)**
> > >
> > > >- Line 341: “Each input data contains the object category label, raw expression, and reasoning from LLM.” what reasoning from LLM? How is it generated? My impression is that this is the input to train the agent. So, we do not have the agent yet to get the reasoning as explained in other parts of the paper. So, this step should be further clarified.
> > > > - Line 341-342: “Then, we manually set the state/action and collect responses as the outputs.” Is the manual annotation done for all 15k input samples? If yes, it should be reported who has done the labeling (i.e., authors themselves or mechanical Turk, etc.), and the details of the instructions given to labelers should be provided.
> > >
> > > We appreciate the suggestions and clarify some points on fine-tuning Real-LOD$^{agent}$:
> > >
> > > 1. The "LLM" in "the reasoning from LLM" refers to the LLM-based reflector defined in Sec. 3.2, not the LLM-based agent. The reflector reasons whether the raw expression matches the target object.
> > > 2. The annotation of the states is manually done for all 15k by ourselves. Our label instructions are that we should reason these states (i.e., 5 states predefined in Sec. 3.2) as the agent based on corresponding input data, especially the "reasoning from LLM", and regular expression is recommended to accelerate the process. After that, following the spirit of LLaVA, the response (including "reasoning" and "actions") is generated by a Vicuna through prompt with several hand-craft in-context examples (shown in Table 11 of the appendix). Then, we manually verify and revise the responses and integrate the input-output pairs into training data following templates as in Fig.5.
> > >
> > > Revised content:
> > > 1. We have modified the unclear presentation to address the confusion in Ln 339-344.
> > > 2. We have provided details of annotation in Ln 341-344 and the LLM prompts used for fine-tuing data generation in the Table 11 of the appendix.
> > >
> > > &nbsp;
> > >
> > > > Line 350: “We randomly select hundreds of filtered expressions and manually check each for a detailed observation”. This is an important detail of the analysis. “Hundreds” should be quantified with exact numbers.
> > >
> > > We appreciate the valuable advice. The exact number is "three hundred", and we have specified this number in Ln 352 (i.e., Ln 350 in origin manuscript).
> > >
> > > &nbsp;
> > >
> > > > Line 379: “we find that the success rate is 74.7% v.s. 35.6%.” these numbers and the experimental setup used to achieve these are not clear.
> > >
> > > Thanks for pointing out this problem. We present details of the success rate and the experimental setup as follows:
> > >
> > > 1. Regarding the success rate, suppose $N$ and $N_s$ represent the number of total expressions and correctly refined expressions, respectively. The success rate can be formulated as $\frac{N_s}{N}$. As presented in Ln 377, the total number of expressions we examined is nearly 11k.
> > > 2. Regarding the experimental setup, we first build a workflow for comparison by replacing the planning step in Real-LOD with a step where one of the five states/actions is randomly selected for further expression refinement. The reflector is used in both workflows to identify whether the final expression refinement is successful. As presented in Ln 379, we set the maximum round for these two workflows to 3.
> > >
> > > The results (i.e., 74.7% v.s. 35.6%) indicate that the reasoning ability of the agent is crucial to our workflow. We have included the above explanation in Sec. 3.3.
> > >
> > > &nbsp;
> > >
> > > > it is mentioned in the abstract that the method results in "50% improvement". Does it mean that the proposed method improved the performance by 50 percentage points? that does not seem to be the case in the experiments.
> > >
> > > We justify that we improve the sota results by 50% upon their own performance relatively rather than the absolute percentage values shown in the benchmark. Taking AP-des in OmniLabel-All Benchmark as an example, our method (i.e., 36.5) outperforms the 150% of the sota method FIBER (i.e., 22.3 x 150% ≈ 33.5).
> > >
> > > &nbsp;
> > >
> > > > It is also more informative if the authors report the average improvement rather the maximum.
> > >
> > > Thanks for this valuable advice. We follow the same experimental setting of [1,2]. The average improvement is not specified in the main text because the performance fluctuations of the LOD benchmarks (i.e., OmniLabel, DOD, RefC/+/g, and OVDEval) are within a negligible range.
> > >
> > > [1] Zhao et al. An Open and Comprehensive Pipeline for Unified Object Grounding and Detection. arXiv, 2024.
> > >
> > > [2] Li et al. GLIP: Grounded Language-Image Pre-training. CVPR, 2022.

---

> > > > ### Author Response · Authors · 2024-11-23
> > > > **Response to Reviewer LXzJ (4/5)**
> > > >
> > > > > - It might be easier to understand if section 3.3 (DATA ANALYSIS OF LANGUAGE AND VISUAL OBJECTS) comes before section 3.2 (AGENTIC WORKFLOWS FOR LANGUAGE EXPRESSION RE-ALIGNMENT)
> > > > > - Similarly, it might be helpful to have section 4.2 ( COMPARISONS WITH STATE-OF-THE-ART LOD METHODS) before section 4.1 (ABLATION STUDY)
> > > >
> > > > We appreciate the valuable suggestion. However, we feel that putting 3.3 before 3.2 is incoherent to illustrate our algorithm. The reason is that Sec 3.1 describes the background and proposal of our algorithm, so it is natural to introduce its overall architecture in the following Sec 3.2. In addition, Sec 3.3 not only introduce some settings but also analyzes the effectiveness of our algorithm and the motivation of action design mentioned in Sec 3.2 (Ln 278-295). It may cause confusion if we directly analyze algorithm results without introducing the algorithm first. As for Sec 4.2 and Sec 4.1, we have modified the order of these two sections following the advice.
> > > >
> > > > &nbsp;
> > > >
> > > > > The authors claim significant boosts in performance but the experiments are very limited (only three datasets). More extensive experiments are needed to show the merits of the proposed method.
> > > >
> > > > We would like to clarify that we do not only use three datasets. In Sec. E of the appendix, we have provided two extensive experiments to show the merits of the proposed method:
> > > >
> > > > 1. Comparisons with state-of-the-art LOD models on OVDEval benchmark
> > > > 2. Application to other LOD models on OmniLabel benchmark.
> > > >
> > > > In total, we evaluate our method on four advanced and popular LOD benchmarks (i.e., OmniLabel, DOD, RefC/+/g and OVDEval) containing challenging, flexible and diverse object expressions, which can effectively verify the VL (vision-language) alignment ability of LOD models. The significant improvement in performance on these benchmarks intensely demonstrates the effectiveness of our method. We have added a reference to these experiments in Sec. 4.2 of the revised manuscript.
> > > >
> > > > &nbsp;
> > > >
> > > > > For section 3.3, although authors have done the analysis that shows choosing states and actions by their agent achieves better performance than random selection of states/actions. It is important to also report the accuracy of the agent in choosing the corresponding state/action for each input sample.
> > > >
> > > > Thanks for this valuable advice. The following table reports the agent's accuracy in choosing the corresponding state/action on the validation set of fine-tuning data (1k samples). The results demonstrate that our agent can accurately reason the state/action, especially for the "Wrong" and "Uncertain" states. In addition, the relatively lower accuracy of the "Correct" state indicates that our agent is strict with the quality of expression to prevent the hallucinations as much as possible. We have provided the following table in Sec. J of the appendix.
> > > >
> > > > |          State/Action          |  Acc  |
> > > > |:------------------------------:|:-----:|
> > > > |          Correct/Stop          | 93.1% |
> > > > |           Wrong/LLM            | 99.4% |
> > > > |     Uncertain/Object Crop      | 99.6% |
> > > > | Uncertain/Extended Object Crop | 95.8% |
> > > > |      Uncertain/Highlight       | 90.0% |

---

> > > > > ### Author Response · Authors · 2024-11-23
> > > > > **Response to Reviewer LXzJ (5/5)**
> > > > >
> > > > > > Correct me if I am wrong, but it seems like the agent is only trained on data built from Objects365. It would be interesting to evaluate the model on Objects365 dataset exclusively to analyze how transferable the agent is to other datasets. Although object365 is included in Omnilabel but exclusive numbers for Objects365 help better understand the impact of distribution shift from training to inference.
> > > > >
> > > > > We train the agent based on the data only from Objects365. However, we believe our agentic workflow still retains a high transferability to other datasets because the impact of fine-tuning data sources is within a negligible range. The following two points are our analysis:
> > > > >
> > > > > 1. The fine-tuning data is not directly from the Object365 training set. We convert the image content into a pure language caption and leverage its category to generate raw expressions. After that, to emulate the planning task of the agent in our workflow, we manually reorganize and relabel it into an entirely new data form. The details of this process are shown in Sec. 3.3. After reannotation, the impact of fine-tuning data resources is significantly reduced.
> > > > > 2. Our agentic workflow not only contains the fine-tuned agent but also includes several foundation models (i.e., Vicuna, Llava) for a diverse perspective. These models are leveraged to interact, examine, and refine language expressions iteratively within the workflow. These interactions advance the robustness of our workflow and reduce the impact of data sources used to fine-tune the agent.
> > > > >
> > > > > In addition, we would like to clarify that the Object365 included in OmniLabel is sufficient to demonstrate the impact of the distribution shift from training to inference as original Object365v2. The reason is that the data size of OmniLabel-Object365 (i.e., 6.1k images with 7.8k descriptions) is similar to the original Object365v2 val set (i.e., 8k images with 365 categories). Besides, the OmniLabel-Object365 contains a large number of object descriptions which are more complex and longer than the categories of the original Object365v2. The improvement of performance achieved on other benchmarks (i.e., OmniLabel-COCO, OmniLabel-OI, DOD, RefC/+/g and OVDEval), which is similar to improvement on the OmniLabel-Object365 demonstrates the high transferability of our workflow.
> > > > >
> > > > > &nbsp;
> > > > >
> > > > > > - Why use an LLM as the agent when the input contains both image and text? Isn’t a language-generative VLM a better choice?
> > > > > > - Similarly, in state2/action2, why use an LLM to write a new object description based on the existing caption rather than using a VLM to write the description based on both the image and the caption?
> > > > >
> > > > > Thanks for the valuable questions. The corresponding answers are as follows:
> > > > >
> > > > > 1. We use an LLM as the agent because we empirically find that LLM reasons are more accurate in pure language form, especially when facing long text prompts and cases requiring strong logic. In contrast, the language-generative VLM can not effectively reason when perceiving language and image data simultaneously. A showcase of this ineffectiveness is the model hallucination that produces incorrect language expressions analyzed in Ln 355-365. This ineffectiveness is due to the naturally inadequate alignment between image and language data used for VLM training (e.g. it is difficult to fully represent what an image conveys in 1-2 sentences). Therefore, we choose LLM with pure language form as the agent to leverage its strong logical reasoning ability for accurate and robust planning within our workflow.
> > > > > 2. Similarly, in state2/action2, the rewritten expressions from LLM have fewer hallucinations and are more diverse than those from VLM. Therefore, we use an LLM to write a new object description rather than a VLM.
> > > > >
> > > > > Moreover, we observe that the reflection module, also an LLM, never introduces new errors and has a high recall for Wrong/Uncertain expressions/phrases. This implies that LLM can provide more stable and accurate answers when dealing with text analysis tasks.

---

> > > > > > ### Comment · Reviewer_LXzJ · 2024-11-27
> > > > > >
> > > > > > Thank you for your thorough response! I have updated my final score!

---

### Official Review · Reviewer_J3CB · 2024-11-03

**Soundness:** 3
**Presentation:** 3
**Contribution:** 3
**Rating:** 6
**Confidence:** 4

**Summary:**

The paper presents an agentic approach to improving text-image data for language-based object detection. The agent has access to tools (LLMs and VLMs) and performs planning and reflection steps. The proposed agentic pipeline is used to re-align text-image data and train a model that significantly outperforms prior works.

**Strengths:**

1. The proposed data filtering and processing mechanism improves the data, and leads to better models
2. The method is evaluated on several datasets and the improvements are consistent
3. The method relies on open-source LLMs and VLMs, and the prompts used have been provided, so the work could be reproduced

**Weaknesses:**

1. The agent has access to a variety of actions to choose from (from the states it can be) and there is some intuition that the 6 mis-alignment reasons (category, attribute, etc) will each be beneficial for some of these failure modes. It would be good to see if the used actions and tools actually reflects that intuition with some analysis on which actions were chosen when different mistakes were made
2. Further to the previous point, the agentic pipeline as is now has quite a lot of components. It might be very beneficial if any of them can be removed without significant loss in performance.
3. The paper several times mentions the model trained is a "prevalent" model -- can this be backed up with other works that use the same model? Comparing to these more explicitly would make the contribution of this work stronger
4. The training of the agent is not properly explained. Some things that should be there are 1) how many samples were used to train the agent? It says that these were manually created by the authors using the outputs of the various tools. Did this include all the recurrent steps?

Minor: Section 4.1 introduces forms A,B,C which are then not references in the table and is a bit confusing

**Questions:**

-

---

> ### Author Response · Authors · 2024-11-23
> **Response to Reviewer J3CB (1/2)**
>
> Dear Reviewer J3CB,
>
> We would like to thank you for providing valuable comments, and we answer the concerns raised below. We have included all discussion results in our revised manuscript (content in orange color).
>
> > The agent has access to a variety of actions to choose from (from the states it can be) and there is some intuition that the 6 mis-alignment reasons (category, attribute, etc) will each be beneficial for some of these failure modes. It would be good to see if the used actions and tools actually reflects that intuition with some analysis on which actions were chosen when different mistakes were made
>
> We appreciate the valuable suggestion. In our paper, we have provided several examples to analyze which states/actions were chosen when different misalignments may happen. The following are some of them (omit state 1 and state 2):
>
> 1. When Real-LOD is in state 3, uncertain about the category or attribute of the target object, it plans to execute action 3. This action involves the `object crop` operation, which directly crops the target object for the VLM, and then VLM can better re-percept focusing on the target without introducing other noise. In addition, the question prompt will be redesigned so that VLM can focus on the target object itself.
>     - Fig. 14 in the appendix shows an example of the first cycle. The expression describes the attribute (i.e., color) of the `'dining table.'` For examination of correctness, Real-LOD crops the table and redesigns a specific question prompt that `'Questions:1.What is the color of the table?2. Is the table made of wood or another material?'` for VLM. Then, Real-LOD can acquire information that `'1. The color of the table is brown.2. The table is made of wood.'`
> 2. When Real-LOD is in state 4, uncertain about how the target object relates to its surroundings or accessory objects, it plans to execute action 4. This action involves the `extended object crop` operation, which provides a more extensive region covering the target objects with surroundings for the VLM, and then VLM can better re-percept rather than a simple object crop or a whole image. In addition, the question prompt will be redesigned so that VLM can focus on the relationship between the target object and its surroundings.
>     - Fig. 5 shows an example in the first cycle. The expression describes the horse and its relation to the surroundings. For examination of correctness, Real-LOD crops a larger region of horse and redesigns a specific question prompt that `'Questions:1. Is the horse walking on sand or on a dirt path?'` for VLM. Then, Real-LOD can acquire information that `'the horse is walking on a dirt path, not on sand.'`
> 3. When Real-LOD is in state 5, uncertain about how the target object relates to the whole image, it plans to execute action 5. This action involves highlighting the object region with a red rectangle, and then VLM can better re-percept the location or relationship/interaction with the further or larger objects in the whole image. In addition, the question prompt will be redesigned so that VLM can focus on the relationship of the target object further.
>     - Fig. 16 shows an example in the second cycle. The expression describes the bus and its relation to the whole image (i.e., location). For examination of correctness, Real-LOD highlight the bus without cropping the image and redesigns a specific question prompt that `'Questions: 1.Is the bus on the right side of the image?'` for VLM. Then, Real-LOD can acquire information that `'Yes, the bus is on the right side of the image.'`
>
> As demonstrated in Fig. 7, Real-LOD achieves high success rate of expression refinement in 6 aspects, especially: `Relation` (relation to surrounding objects, such as `apple on table`), `Accessory` (relation to accessory objects, such as `person with a hat`), `Location` (relation to whole image, such as `bus on the right side of image`) and `Behavior` (relation to further objects, such as `man playing frisbee while a group of people watch`).

---

> ### Author Response · Authors · 2024-11-23
> **Response to Reviewer J3CB (2/2)**
>
> > Further to the previous point, the agentic pipeline as is now has quite a lot of components. It might be very beneficial if any of them can be removed without significant loss in performance.
>
> We appreciate the valuable suggestion. We argue that there are not quite a lot of components in the proposed workflow, and each component is significant because our design principle considers all the components as a whole pipeline. In addition, the computational cost of our workflow is affordable and performs off-line without affecting the LOD model inference, as shown in Sec. 4.3.
>
> Besides, we conduct a more detailed ablation study of our agentic workflow. The evaluation results of the success rate are as follows. The details of the experimental setup can be found in Sec. 3.3. In the following table, the "w/o Planning" is the same as the random selection schema in Sec. 3.3. "w/o Cyclic Workflow" indicates the workflow with only one cycle. The results intuitively illustrate the importance of each component to our agentic workflow.
>
> We have concluded the following table in Sec. J of the appendix. This question will also motivate us to explore building a more lightweight workflow in the future.
>
> |       Method        | Success Rate |
> |:-------------------:|:------------:|
> |      Real-LOD       |    74.7%     |
> |    w/o Planning     |    35.6%     |
> |    w/o Action 2     |    18.0%     |
> |    w/o Action 3     |    53.0%     |
> |    w/o Action 4     |    51.8%     |
> |    w/o Action 5     |    57.4%     |
> | w/o Cyclic Workflow |    60.7%     |
>
> &nbsp;
>
> > The paper several times mentions the model trained is a "prevalent" model -- can this be backed up with other works that use the same model? Comparing to these more explicitly would make the contribution of this work stronger
>
> We appreciate the valuable suggestion. The prevalent model can be backed up with other works. However, to our knowledge, no existing methods directly related to ours improve the VL (vision-language) alignment ability with agentic workflow from a data-centric perspective. We have considered this problem and performed various experiments in Sec. 4 and Sec. E to demonstrate the effectiveness of our workflow. Especially in the ablation study of Sec. 4, we conduct three data configurations (i.e., A, B, and C forms) to train the "prevalent" model. In A form, all the generated data pairs are used for training, which can serve as the application of previous work[1]. This work aims to diversify the object expressions. The 3% improvement of AP-des in OmniLabel-COCO brought by C forms, where we use the re-aligned data pairs via our workflow for training, is strong evidence to demonstrate the effectiveness.
>
> [1] Dang, et al. InstructDET: Diversifying Referring Object Detection with Generalized Instructions. ICLR 2024.
>
> &nbsp;
>
> > The training of the agent is not properly explained. Some things that should be there are 1) how many samples were used to train the agent? It says that these were manually created by the authors using the outputs of the various tools. Did this include all the recurrent steps?
>
> We appreciate the valuable suggestions and make some clarification on agent fine-tuning here:
>
> 1. We use 15k samples to train the agent, as shown in the first paragraph of Sec. 3.3.
> 2. The training data do not include all the recurrent steps. This is because the planning process of the agent in each recurrent step is the same and we only need to focus on the behavior of agent in one recurrent step.
>
> We have provided more details about agent fine-tuning in the first paragraph of Sec. 3.3.
>
> &nbsp;
>
> > Minor: Section 4.1 introduces forms A,B,C which are then not references in the table and is a bit confusing
>
> Thanks for pointing out this problem. We have added references to the A, B, and C forms in Table 4 (i.e., origin Table 1).

---

### Author Response · Authors · 2024-11-23
**Response to AC and All Reviewers**

Dear AC and all reviewers:

We sincerely appreciate your time and efforts in reviewing our paper. We are glad to find that reviewers recognized the following merits of our work:

- **Novelty and soundness ([LXzJ, M1hG]):** We propose an agentic workflow comprising planning, tool use, and reflection steps as a whole pipeline, with well-designed states/actions, to improve the VL (vision-language) alignment from a data-centric perspective. This idea is novel and well-motivated.
- **Impressive performance ([J3CB, e1U7, M1hG]):** With the proposed Real-LOD, the retrained LOD model achieves sota performance across challenging benchmarks, underscoring the practical efficacy of this approach.
- **Sufficient and reasonable experiments ([J3CB, LXzJ, e1U7, M1hG]):** The ablation study and experimental validations are sufficient to verify the effectiveness of Real-LOD in improving the VL alignment quality.
- **Well-designed visualizations and tables ([e1U7, M1hG]):** The figures and tables in the papers are clear and informative, enhancing the reader's understanding of the methodology.

We also thank all reviewers for their insightful and constructive suggestions, which help further improve our paper. In addition to the pointwise responses below, we summarize the major revision in the rebuttal according to the reviewers' suggestions.

- **Additional experiments ([J3CB, LXzJ]):** We conduct a more detailed ablation study of our agentic workflow. The evaluation results of the success rate are added in the Sec. J of the appendix, which demonstrates the importance of each component to our workflow ([J3CB]). In addition, we report the accuracy of the agent in choosing the corresponding state/action on the validation set of fine-tuning data in Sec. J of the appendix. This indicates that our agent in the proposed workflow can accurately reason the state/action ([LXzJ]).
- **Manuscript update ([J3CB, LXzJ, M1hG]):** We modify manuscript presentations, including grammar error correction and clearifying notifications ([J3CB, LXzJ]). We also provide detailed information on raw expression generation in Sec 3.1 ([LXzJ]), agent input and behavior in Sec. 3.2 ([LXzJ]), experimental setup for success rate computation ([LXzJ]) and agent fine-tuning in Sec. 3.3 ([J3CB, LXzJ]). Besides, we reorganize Sec 3.3, place Sec 4.1 behind Sec 4.2 ([LXzJ]) and replace the naming that is not easy to distinguish ([M1hG]). We also add references to the corresponding appendix content in the main text. In the appendix, we add a detailed explanation for the expression generation pipeline in Sec. A ([LXzJ]). We also provide prompts for raw expression ([LXzJ]) and fine-tuning data generation ([J3CB, LXzJ]) in Sec. G. The prompts for LLM-for-Rewriting and LLM-Reflector are also updated to be more apparent in Sec. G ([J3CB, LXzJ]). In addition, we include visualization of some failure cases and corresponding analysis in Sec. I ([M1hG]). Our revision is marked in orange.

Our detailed responses can be found in the following. We hope our responses can clarify the confusion and address the raised concerns. Again, we sincerely thank all reviewers for their efforts and time.

&nbsp;

Best regards,

The Authors

---

### Author Response · Authors · 2024-12-04
**Summarization of author-reviewer discussion and our work contributions**

Dear AC and all reviewers:

First of all, we thank AC for organizing the review in smooth progress, where we have communicated well with reviewers. Also, we appreciate the reviewers making an effort to provide constructive comments to improve our submission. In general, we thank reviewers for the acknowledgement, including novelty ([LXzJ, M1hG]), impressive performance ([J3CB, e1U7, M1hG]), sufficient experiments ([J3CB, LXzJ, e1U7, M1hG]), and well-designed figures and tables ([e1U7, M1hG]).

During the discussion phase, we have sufficiently addressed all the concerns raised by LXzJ, which is acknowledged in LXzJ's last comment. On the other hand, we have well addressed the majority concerns raised by e1U7, leaving e1U7's last comment regarding the writing.

We have carefully improved our writing according to the constructive comments raised by all reviewers ([LXzJ, e1U7, J3CB, M1hG]). Our improvement is conducted by point-to-point clarifications in the revised manuscript (shown in orange color), including:

- Overall manuscript writing. We correct grammar errors and clarify unclear notifications ([J3CB, LXzJ]). We also replace the ambiguous notation, e.g., replace Real-LOD$^{agent}$ with Real-Agent ([M1hG]).
- In Sec 3.1, we provide detailed information on raw expression generation. We offer more information and incorporate the prompts for this process in Sec. A and Sec. G of the appendix, respectively.
- In Sec. 3.2, we elucidate agent input and provide more details about our three stages (i.e., planning, tool use, and reflection). We also clearly show the prompts for LLM-for-Rewriting and LLM-Reflector in Sec. G of the appendix ([LXzJ]).
- In Sec. 3.3, we clarify the experimental setup for success rate computation ([LXzJ]) and provide more details of agent fine-tuning in Sec. 3.3 ([J3CB, LXzJ]). We also include the prompts used for agent fine-tuning in Sec. G of the appendix ([J3CB, LXzJ]). In addition, we also reorganize Sec. 3.3 by three subsections ([LXzJ]).
- We place Sec 4.1 behind Sec 4.2 and add references to additional experiments on other benchmarks in Sec. E of the appendix ([LXzJ]). In addition, visualizations of some failure cases and corresponding analyses are included in Sec. I ([M1hG]).
- Moreover, we conduct a more detailed ablation study to demonstrate the importance of each component in our workflow ([J3CB]) and provide additional evaluation results to verify our method's effectiveness further ([LXzJ]). We also respond to the questions regarding analysis of states/actions([J3CB]), the reason to use LLM rather than VLM ([LXzJ]), how our workflow to solve the hallucinations problem ([e1U7]), choice of VLM ([M1hG]), the influence of without agent fine-tuning ([M1hG]), lite version ([M1hG]).

**Summarization of our work contributions**: We pioneerly propose an agentic workflow by designing reasoning/planning, tool use, and reflection steps as a whole pipeline to improve VL alignment from a data-centric perspective. Motivated by the neural-symbolic spirit, we propose 5 states/actions based on our data analysis regarding misaligned visual objects and languages. These actions, controlled by our agent, facilitate VLMs to re-perceive ROI areas for re-alignment, where our agent proposes customized questions by itself in the VQA form to address each mis-alignment issue specifically. We believe our pioneering work will motivate huge investigations on designing agentic workflows to solve computer vision oriented tasks, bringing a new era of advancing the VL field by using AI agents.

All the raised issues are carefully addressed in the rebuttal, discussions, and revised manuscripts. We sincerely hope the AC/SAC to have a check of all the materials, and render a convincing decision. Finally, we wish ICLR 2025 a great success.

&nbsp;

Best regards,

The Authors

---

### Meta-Review · Area_Chair_LEyG · 2024-12-20

**Metareview:**

The paper main-contribution is a method to iteratively improve synthetically generated for training language-based object detection (LOD). Through multiple rounds of generation and reflection steps using "tools" (LLMs and VLMs). Using this generated data for training, the language-based detector performs well on language based object detection task.

The reviewers praised the paper for the innovative data generation approach, and also agreed that the performance improvements are consistent. Both LXzj and E1U7, during the rebuttal and reviewers discussion found the writing quality below par. The authors addressed several writing related comments from the reviewers, but there is a need to incorporate them in the main manuscript more carefully.

While there were concerns raised about the novelty of the proposed workflow, the quantitative improvements on the LOD downstream task, as well as careful design choices throughout the workflow will be useful to the language-based object detection community. We recommend authors to discuss lessons learned during designing the workflow in the manuscript to make the paper broadly appeal to the wider community.

**Additional Comments On Reviewer Discussion:**

There were several questions raised about experimental design, agent design, etc that saw detailed response from the authors.

Experiments on Additional Dataset: Reviewer LXzj asked for additional experiments to show the merit of the proposed method. The authors clarified that there are additional experiments in appendix E (including OmniLabel, DOD, RefC/+/g and OVDEval).

Accuracy of agent choosing state/action fro each input sample: The authors performed additional experiments and showed high accuracy while choosing state/action.

Reason for choosing LLMs as the agent: The authors clarified that, from their experiments they found LLMs to produce less hallucination, and more accurate results. We encourage the authors to present this empirical comparison in the supplementary.

Error Analysis of the workflow: The reviewers asked for error analysis of the workflow, and the authors responded by adding those in the appendix.

---

### Decision · Program_Chairs · 2025-01-22

Accept (Poster)